# Diff3DS: Generating View-Consistent 3D Sketch via Differentiable Curve Rendering

**Yibo Zhang**[1]    **Lihong Wang**[1]    **Changqing Zou**[2,3]    **Tieru Wu**[1,4]    **Rui Ma**[1,4]*
[1]Jilin University    [2]State Key Lab of CAD&CG, Zhejiang University    [3]Zhejiang Lab
[4] Engineering Research Center of Knowledge-Driven Human-Machine Intelligence, MOE, China
`{ybzhang23, wlh23}@mails.jlu.edu.cn`
`{wutr, ruim}@jlu.edu.cn`
`aaronzou1125@gmail.com`

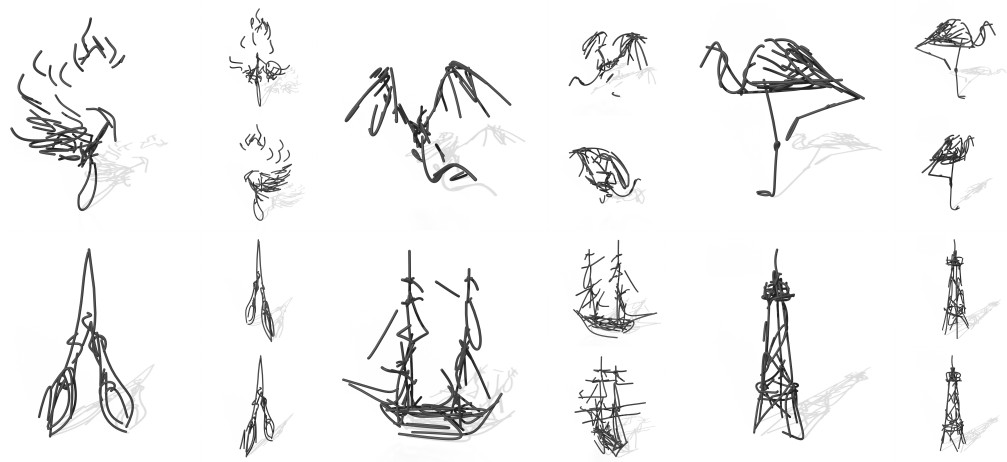

Figure 1: In this paper, we propose Diff3DS, a novel differentiable rendering framework for generating view-consistent 3D sketch from flexible inputs such as a single image or text.

## Abstract

3D sketches are widely used for visually representing the 3D shape and structure of objects or scenes. However, the creation of 3D sketch often requires users to possess professional artistic skills. Existing research efforts primarily focus on enhancing the ability of interactive sketch generation in 3D virtual systems. In this work, we propose **Diff3DS**, a novel differentiable rendering framework for generating view-consistent 3D sketch by optimizing 3D parametric curves under various supervisions. Specifically, we perform perspective projection to render the 3D rational Bézier curves into 2D curves, which are subsequently converted to a 2D raster image via our customized differentiable rasterizer. Our framework bridges the domains of 3D sketch and raster image, achieving end-to-end optimization of 3D sketch through gradients computed in the 2D image domain. Diff3DS can enable a series of novel 3D sketch generation tasks, including text-to-3D sketch and image-to-3D sketch, supported by the popular distillation-based supervision, such as Score Distillation Sampling. Extensive experiments have yielded promising results and demonstrated the potential of our framework. Project page is at `https://yiboz2001.github.io/Diff3DS/`

## 1 Introduction

3D sketches, which utilize strokes to emphasize abstraction and visually encapsulate the 3D shape and structure of objects or scenes, serve as indispensable tools for visualizing concepts and ideas. Existing research efforts have primarily focused on exploring how to enable interactive creation of 3D sketch

---

*Corresponding author.

in 3D virtual systems (Wesche & Seidel, 2001; Grimm & Joshi, 2012; Bae et al., 2008) and immersive environments (Yu et al., 2021a; Xin et al., 2008; Kim et al., 2023; Jiang et al., 2021; Kim & Bae, 2016; Yu et al., 2021b; Arora & Singh, 2021; Yue et al., 2017). Such interactive 3D sketch creation tools provide artists with unprecedented freedom, allowing them to draw their inspirations directly in the immersive 3D environment. However, it usually requires users to possess professional artistic skills and experience for 3D operation, making the 3D sketch creation less friendly for ordinary users. There is a notable absence of a user-friendly *3D sketch generation* method in the community. Moreover, in the area of wire art generation, 3D sketch or 3D curves are also widely studied to abstract the desired visual concepts from diverse inputs, e.g., 3D surfaces (Yang et al., 2021), multi-view images (Liu et al., 2017) and text (Tojo et al., 2024; Qu et al., 2024). These representations provide essential shape and structure of the artwork as well as hold significant potential as intermediate formats for tasks such as 3D reconstruction. However, generating view-consistent 3D sketches from flexible inputs, such as text or a single image, remains largely unexplored.

In recent years, content reconstruction or generation via differentiable rendering has gained great attention in the fields of computer graphics and vision research. The goal is to convert the traditional rendering pipeline into a differentiable image synthesis process and employ the 2D image supervision to optimize the object or scene representation. For example, NeRF (Mildenhall et al., 2021) bridges the 3D neural representation and raster image domain with a differentiable volume rendering pipeline, and has contributed to the further development of the 3D generation (Mohammad Khalid et al., 2022; Tsalicoglou et al., 2024; Jain et al., 2022; Poole et al., 2023; Lin et al., 2023) by leveraging the continuous advancements in multimodal supervision (Radford et al., 2021; Poole et al., 2023). For the generation of vectorized or parametric content, differentiable rendering has also been investigated (Li et al., 2020; Schaldenbrand et al., 2022; Frans et al., 2022; Jain et al., 2023; Xing et al., 2023; Qu et al., 2023; Xing et al., 2024; Banerjee et al., 2024; Vinker et al., 2022; 2023; Thamizharasan et al., 2024b;a). One representative work is DiffVG (Li et al., 2020), which proposes a differentiable 2D vector graphics rasterizer to compute gradients from raster images and optimize the vectorized image (e.g., 2D Bézier curves and other parametric shapes) via back-propagation. For generation of sketch, CLIPasso generates 2D sketches using DiffVG to optimize the parameters of the 2D Bézier curves directly with respect to a CLIP-based perceptual loss. Recently, 3Doodle (Choi et al., 2024) generates view-consistent 3D sketches of the target object by directly optimizing the parameters of 3D strokes to minimize the perceptual losses for given multi-view images. It also proposes a differentiable 3D curve rendering pipeline which integrates DiffVG as a key component. Although 3Doodle achieves promising results for generating 3D sketches, its optimization requires supervision from multiview images, which makes it more like a reconstruction pipeline instead of a more flexible generation pipeline. Meanwhile, its performance is constrained by the inherent limitations of DiffVG's original design. It relies on an approximate perspective projection to obtain the 2D sketch rendering and ignores the depth order of the curves, which may lead to color conflicts in the 3D sketch and limit its effectiveness on more complex tasks like colored sketch generation.

In this paper, we present Diff3DS, a novel differentiable rendering framework for generating view-consistent 3D sketch from flexible inputs such as a single image or text. Specifically, we represent the 3D sketch as a set of 3D *rational* Bézier curves and perform the perspective projection to obtain the 2D rational Bézier curves. Then, we propose a new differentiable rasterizer based on DiffVG to accurately render the projected 2D curves based on the depth order, so that more accurate occlusion relationships between curves can be modeled, especially for colored curves. Our framework supports end-to-end optimization of 3D curve primitives under flexible supervisions such as the distillation-based loss. By employing the recent Score Distillation Sampling algorithm to distill prior knowledge from pre-trained 2D image generation model, we achieve 3D sketch generation from the text or single image input. We conduct quantitative and qualitative comparisons between Diff3DS and related methods. The results demonstrate the superiority of our approach in generating view-consistent 3D sketches. Ablation studies further validate the effectiveness of the key components of our method. Additionally, analysis of the rasterizer shows its potential for generating colored sketches.

We summarize our contributions as follows: (1) We propose Diff3DS, a novel differentiable rendering framework for generating view-consistent 3D sketch from flexible inputs such as a single image or text. (2) We are the first to represent the 3D curve as 3D rational Bézier curve and design a depth-aware rasterizer that can enable precise and differentiable rendering of both black and colored 3D curves. (3) We conduct comprehensive experiments on novel text-to-3D and image-to-3D sketch generation tasks and the results demonstrate the superiority of our method.

## 2 RELATED WORKS

**3D Sketch Generation**   Existing research on 3D sketch generation mainly focus on interactive artistic sketch creation within 3D virtual systems (Wesche & Seidel, 2001; Grimm & Joshi, 2012; Bae et al., 2008), and immersive environments (Yu et al., 2021a; Xin et al., 2008; Jiang et al., 2021; Kim & Bae, 2016; Yu et al., 2021b; Arora & Singh, 2021). However, these works usually require users to possess professional artistic skills and 3D operation experience, making it less user-friendly for ordinary users. The recent 3Doodle (Choi et al., 2024) is able to generate expressive 3D sketches from multiview image observations, but it is more like multiview reconstruction instead of generation and the quality of the generated results highly depends on the number of observation viewpoints. By distilling prior knowledge from pre-trained 2D image generation models using methods like Score Distillation Sampling (Poole et al., 2023; Liu et al., 2023), we achieve user-friendly and view-consistent 3D sketch generation from flexible text or single image input.

**Differentiable Rendering of Curves**   There have been several works for differentiable rendering of curves. DiffVG proposes a differentiable rasterizer for the creation of vector graphics represented by 2D curves and shapes. DRPG (Worchel & Alexa, 2023) investigates differentiable rendering of 3D parametric curves and surfaces, by piecewisely approximating the continuous parametric representations with a triangle mesh. 3Doodle (Choi et al., 2024) shares a similar differentiable rendering pipeline with us, including projecting the 3D Bézier curves to 2D Bézier curves, which are then rendered by the differentiable rasterizer from DiffVG. However, 3Doodle approximates the perspective projection as an orthographic projection to obtain 2D projected curves and ignores the depth ordering between the curves. Instead, we perform perspective projection to the 3D rational Bézier curves and designs a new DiffVG-based ratersizer which can maintain the depth ordering between the projected 2D rational Bézier curves.

**Multimodal-Driven 3D Content Generation**   Motivated by the success of the text-to-image research, pioneering works (Poole et al., 2023; Wang et al., 2023a) introduce the Score Distillation Sampling (SDS) algorithm and leverage the pre-trained text-to-image models as prior knowledge to optimize 3D representations. Subsequent works combine SDS with various differentiable 3D representations to explore the capabilities of text-to-3D object generation, such as DMTet (Lin et al., 2023; Chen et al., 2023) and 3D Gaussians (Tang et al., 2024; Yi et al., 2024). Furthermore, Zero-1-to-3 (Liu et al., 2023) proposes an image-guided SDS algorithm that distills the 3D-consistent priors to optimize 3D representations given the input image, and its pipeline inspires the development of subsequent image-to-3D object methods (Liu et al., 2023; Qian et al., 2024; Wang & Shi, 2023; Wu et al., 2023; Sta, 2023). Distinguishing from prior endeavors, our work stands out by focusing on generating view-consistent 3D sketch by optimizing 3D parametric curve primitives.

## 3 PRELIMINARY

DiffVG (Li et al., 2020) proposes a differentiable rasterizer for vector graphics that supports converting 2D Bézier curves to the raster image domain and back-propagating the gradient for optimization. Given the 2D curves parameters $\Theta$, the vector graphics scene is defined as $f(x, y, \Theta)$, and the raster image is defined as the 2D grid sampling over the space of $f(x, y, \Theta)$. To compute the color of a 2D location $(x, y) \in \mathbb{R}^2$, DiffVG utilizes the inside-outside test (Neh) to find the curves overlapped with the location initially. Subsequently, it sorts them according to a user-specified order and calculates the color using alpha blending (Porter & Duff, 1984). Due to the inside-outside test, the scene function $f$ is not differentiable with respect to curve parameters. Thus, it uses the anti-aliasing technique to make the pixel color differentiable. By *prefiltering* $f$ over a convolution kernel $k$ with support $A$, sampling $I(x, y)$ at pixels can yield an alias-free image:

$$I(x, y) = \iint_A k(u, v) f(x - u, y - v; \Theta) \, \mathrm{d}u \, \mathrm{d}v. \tag{1}$$

Due to the integration over the filter support region, the focus shifts from the color value at the center point to the average pixel color. The continuous change in the average color induced by the curve movements makes the function $I(x, y)$ differentiable.

The goal is to compute the gradients of $I$ with respect to $\Theta$:

$$\frac{\partial I(x,y)}{\partial \Theta} = \frac{\partial}{\partial \Theta} \iint_A k(u,v) f(x-u, y-v; \Theta) \, \mathrm{d}u \, \mathrm{d}v. \tag{2}$$

DiffVG proposes two approaches *Monte Carlo Sampling* and *Analytical Prefiltering* to evaluate the pixel integral, which does not have a closed-form solution in general. We focus on the first approach which performs better. The pixel integral can be discretized with Monte Carlo sampling (Ganacim et al., 2014):

$$I(x,y) = \iint_A k(u,v) f(x-u, y-v; \Theta) \, \mathrm{d}u \, \mathrm{d}v \approx \frac{1}{N} \sum_i^N k(u_i, v_i) f(x-u_i, y-v_i; \Theta), \tag{3}$$

but geometric discontinuities hinder the interchangeability of the integral and differential operators, which consequently obstructs the direct representation of $\frac{\partial I}{\partial \Theta}$ as a discretizable integral. Thus, it rewrites the pixel color as the sum of integrals over multiple sub-regions $A_i$, and applies the Reynolds transport theorem to handle all the discontinuous changes on the boundary:

$$\frac{\partial I(x,y)}{\partial \Theta} = \sum_i \iint_{A_i(\Theta)} \frac{\partial}{\partial \Theta} g(u,v) \, \mathrm{d}u \, \mathrm{d}v + \sum_i \int_{\partial A_i(\Theta)} \left( \frac{\partial p(t)}{\partial \Theta} \cdot n(t) \right) g(p(t)) \, \mathrm{d}p(t), \tag{4}$$

where $g(u,v)$ is the multiplication of the scene function $f$ and kernel $k$ for brevity, $\partial A_i$ is the boundary of area $A_i$, $n(t)$ is the outward normal of $\partial A_i$, and $p(t)$ denotes the 2D points on the $\partial A_i$. The first integral is responsible for the differentiation of color and the transparency, while the second term is responsible for the change of the boundaries, and the boundary integral can be estimated by the Monte Carlo estimator as:

$$\sum_i \int_{\partial A_i(\Theta)} (\nabla_\Theta p \cdot n) g(p(t)) \, \mathrm{d}p(t) \approx \frac{1}{N} \sum_j \frac{(\nabla_\Theta p_j \cdot n_j) \left( g(p_j + \epsilon n_j) - g(p_j - \epsilon n_j) \right)}{P(p_j|c) P(c)}, \tag{5}$$

where $\epsilon$ is a small number and $P(p_j|c)P(c)$ denotes the probability density of sampling the curve $c$ and point $p_j$ on the boundaries.

## 4 DIFF3DS

### 4.1 OVERVIEW

We propose Diff3DS, a novel differentiable rendering framework for generating view-consistent 3D sketch by optimizing 3D parametric curves. A 3D sketch $\tilde{\Theta}$ is defined as a collection of 3D parametric curve strokes that reside in 3D space. We particularly focus on the 3D rational Bézier curve and our derivation supports the 3D linear, quadratic, and cubic rational Bézier curves. The theoretical trainable parameters include the control point position, curve width, and rgb-alpha color. Follow the previous methods (Vinker et al., 2022; 2023; Choi et al., 2024), we represent each stroke by a 3D cubic Bézier curve and only optimize the control point position in our current implementation. Moreover, weights for rational Bézier curves are disregarded to simplify the computation. Fig. 3 illustrates the pipeline.

Diff3DS is designed with three stages. First, to render the 3D rational Bézier curve strokes $\tilde{\Theta}$ at a given viewpoint, we perspectively project these curves onto the camera plane (Sec. 4.2). Then, we render the projected 2D rational Bézier curves $\Theta$ to the raster image using a customized differentiable rasterizer. (Sec. 4.3). Finally, we will perform back-propagation with the gradient computed from the rendered image to optimize the 3D sketch parameters (Sec. 4.4).

### 4.2 3D RATIONAL BÉZIER CURVE PROJECTION

In this section, we primarily explain how to project the 3D rational Bézier curves in camera coordinates onto the 2D camera focal plane, and detailed proof is provided in Appendix B. Given the control points $\tilde{P}_i = (\tilde{P}_i^x, \tilde{P}_i^y, \tilde{P}_i^z) \in \mathbb{R}^3$, the associated non-negative weights $\tilde{w}_i$ and the Bernstein polynomials $B_{i,n}(t)$, for $0 \le t \le 1$, define the 3D rational Bézier curve $\tilde{p}(t)$ of degree $n$ by:

$$\tilde{p}(t) = \sum_{i=0}^n \frac{B_{i,n}(t) \tilde{w}_i}{\sum_{j=0}^n B_{j,n}(t) \tilde{w}_j} \tilde{P}_i = (\tilde{x}(t), \tilde{y}(t), \tilde{z}(t)). \tag{6}$$

Assume the focal plane is $z = f$, where $f$ is the focal length, the corresponding projected 2D curve $p(t)$ under the pinhole camera perspective projection can be defined as:

$$p(t) = (x(t), y(t)) = (\frac{\tilde{x}(t)}{\tilde{z}(t)}f, \frac{\tilde{y}(t)}{\tilde{z}(t)}f).$$

(7)

Due to rational Bézier curves are projective invariant, the curve $p(t)$ still is identical to the 2D rational Bézier curve defined by projected control points. Given projected 2D control points $P_i = (P_i^x, P_i^y) = (\frac{\tilde{P}_i^x}{\tilde{P}_i^z}f, \frac{\tilde{P}_i^y}{\tilde{P}_i^z}f) \in \mathbb{R}^2$ and adjusted weights $w_i = \tilde{w}_i \tilde{P}_i^z$, $p(t)$ can be written as following:

$$p(t) = \sum_{i=0}^{n} \frac{B_{i,n}(t)w_i}{\sum_{j=0}^{n} B_{j,n}(t)w_j} P_i = (x(t), y(t)).$$

(8)

### 4.3 2D RATIONAL BÉZIER CURVES DIFFERENTIABLE RASTERIZATION

This section primarily focuses on the differentiable rasterization of 2D curves $\Theta$ obtained through projection. To render the projected rational Bézier curves and maintain their occlusion relationship in 3D space, especially for colored curves, we customize a new differentiable rasterizer based on DiffVG (Li et al., 2020). Given a 2D pixel location $(x, y)$ and projected curves $\Theta$, we first employ a inside-outside test to identify all the overlapping curves with this pixel location. Then, we sort the overlapping curve points and calculate the color of this pixel location using alpha blending algorithm.

The inside-outside test identifies a curve as overlapping with a location if the closest distance between them is less than half of the curve width, and the closest point to the location are considered as the overlapping point . However, the identification is challenging. For the $n$th-degree rational curve $p(t)$, computing the $t$ that minimizes the distance $(p(t) - q)^2$ where $q = (x, y)$ equals to solve the roots of a polynomial with degree $3n - 2$. For the cubic rational curve, a 7th order polynomial needs to be solved, which does not have a closed-form solution. Inspired by DiffVG, we solve the polynomial using bisection and the Newton-Raphson method (William H. Press & Flannery., 2007). The iterative solver obtains its initial guess from *isolator polynomials* (Sederberg & Chang., 1994) – the real roots of a 7th order polynomial can be isolated by the roots of two cubic auxiliary polynomials. Please refer to Appendix C for more details.

The rasterized result should faithfully maintains the occlusion in 3D space. In our task, different points along the same projected 2D curve may have different depths. Therefore, each point should be assigned with its own specific order for the color blending process. In the original DiffVG, each curve is initialized with a user-specific order, meaning all the points along the same curve share this uniform order. In contrast, our sort order is determined by the depth order among curve primitives. For each overlapping point, we compute its unprojected z-depth in the 3D space using the corresponding 2D projected curve control points, and employ it as the primary sort order. To mitigate the z-fighting issue [1] caused by floating-point errors, we also utilize a user-specified order as the secondary order. After obtaining the 2D curve scene function $f(x, y, \Theta)$, the rendered raster image $I$ will be calculated using the *Monte Carlo Sampling* strategy (Sec. 3) with Eq. (3). Fig. 2 shows an example. Our rasterizer faithfully renders the colored curves and maintains occlusions according to the depth order.

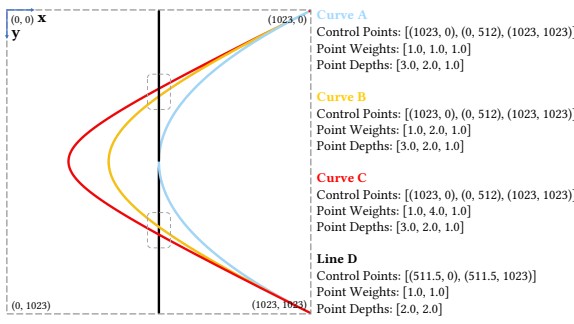

Figure 2: A rasterized result contains 3 quadratic rational Bézier curves and a line. All curves share the same control point positions in pixel space and the depths but with different weights. Our rasterizer faithfully renders the curves and maintains occlusions according to the depth order. (e.g., The upper half of each curve has a greater depth than the line, while the lower half has a lesser depth. This difference results in varying color blending outcomes at the overlapping regions).

---

[1] https://en.wikipedia.org/wiki/Z-fighting

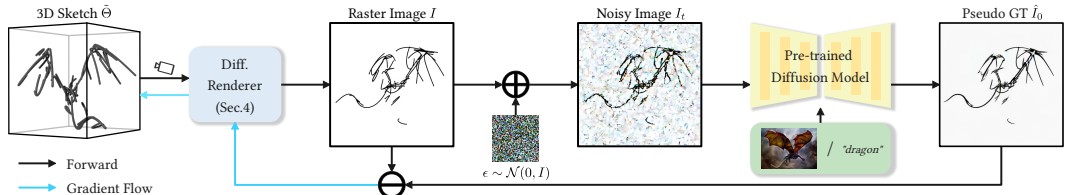

Figure 3: We generate the 3D sketch $\tilde{\Theta}$, represented as a set of 3D strokes, from the text or image input. We render the raster image $I$ from the random camera via our differentiable renderer (Sec. 4). Then, the pre-trained diffusion model, conditioned on the input, diffuses the rendering $I$ and predicts the pseudo ground truth $\hat{I}_0$. The discrepancies between $\hat{I}_0$ and $I$ are used to update the 3D sketch.

### 4.4 GRADIENTS BACK-PROPAGATION

The rendering framework is fully differentiable since the projection and rasterizer are differentiable. Thus, the gradients of $I$ with respect to 3D curves parameter $\tilde{\Theta}$ can be calculated with the chain rule: $\frac{\partial I(x,y)}{\partial \tilde{\Theta}} = \frac{\partial I(x,y)}{\partial \Theta} \frac{\partial \Theta}{\partial \tilde{\Theta}}$, where $\frac{\partial I}{\partial \Theta}$ denotes the gradients of $I$ with respect to 2D projected curves $\Theta$ that can be computed using Eq. (4) and Eq. (5), and $\frac{\partial \Theta}{\partial \tilde{\Theta}}$ denotes the gradients of projected 2D curves $\Theta$ with respect to 3D curves $\tilde{\Theta}$. $\frac{\partial I(x,y)}{\partial \Theta}$ calculates the gradients of control point positions, color, and width for projected 2D curves. The position gradients of 3D curve control points are calculated with $\frac{\partial \Theta}{\partial \tilde{\Theta}}$, which is based on the gradients of the projected 2D control points. The color and width gradients of 3D curves directly inherit the corresponding gradients of the projected 2D curves.

## 5 MULTIMODAL-DRIVEN 3D SKETCH GENERATION

Diff3DS supports end-to-end optimization of 3D sketch under flexible supervisions. By employing the Score Distillation Sampling algorithm, we propose the promising tasks of text-to-3D sketch and image-to-3D sketch, with the goal of generating 3D sketch from flexible text or single image input.

### 5.1 TEXT-TO-3D SKETCH

Pioneering works (Poole et al., 2023; Wang et al., 2023a) propose the Score Distillation Sampling (SDS) algorithm, utilizing the prior of a pre-trained 2D text-to-image model to optimize 3D representations. By integrating SDS with Diff3DS, we distill the diffusion prior to generate the 3D sketch $\tilde{\Theta}$ from the input text. Given a pre-trained diffusion model $\epsilon_\phi$, text embedding $y$, rendered image $I$ with our rasterizer $\mathcal{R}$ and the noise timestep $t$, image $I$ will be added with noise to obtain a noisy image $I_t$: $I_t = \sqrt{\bar{\alpha}_t} I + \sqrt{1 - \bar{\alpha}_t} \epsilon$, where $\epsilon \sim \mathcal{N}(0, I)$ and $\bar{\alpha}_t$ is the cumulative product of scaling at timestep $t$. Then the gradient of SDS loss on $\tilde{\Theta}$ is given by:

$$\nabla_{\tilde{\Theta}} \mathcal{L}_{SDS}(\phi, I = \mathcal{R}(\tilde{\Theta})) = \mathbb{E}_{t,\epsilon} \left[ \omega(t)(\hat{\epsilon}_\phi(I_t; y, t) - \epsilon) \frac{\partial I}{\partial \tilde{\Theta}} \right], \tag{9}$$

where $\omega(t)$ is a weight based on the timestep $t$, and the predicted noise sampled by classifier-free guidance (CFG) (Ho & Salimans, 2021) with weight $\lambda$ denotes as:

$$\hat{\epsilon}_\phi(I_t; y, t) = \epsilon_\phi(I_t; \emptyset, t) + \lambda(\epsilon_\phi(I_t; y, t) - \epsilon_\phi(I_t; \emptyset, t)). \tag{10}$$

By deriving the pseudo ground truth image $\hat{I}_0$ with one-step denoising:

$$\hat{I}_0 = \frac{1}{\sqrt{\bar{\alpha}_t}}(I_t - \sqrt{1 - \bar{\alpha}_t} \hat{\epsilon}_\phi(I_t; y, t)), \tag{11}$$

we can transform Eq. (9) into an equivalent form as follows:

$$\nabla_{\tilde{\Theta}} \mathcal{L}_{SDS}(\phi, I = \mathcal{R}(\tilde{\Theta})) = \mathbb{E}_{t,\epsilon} \left[ \omega(t) \frac{\sqrt{\bar{\alpha}_t}}{1 - \sqrt{\bar{\alpha}_t}} (I - \hat{I}_0) \frac{\partial I}{\partial \tilde{\Theta}} \right], \tag{12}$$

which can be regarded as matching the input view $I$ with predicted $\hat{I}_0$.

Similar with (Poole et al., 2023), due to the bias of 2D text-to-image model, we encounter serious view inconsistency problem known as the Janus problem. This problem persists even when directional prompts (e.g., "the front view of...") are used to distinguish between different views. To mitigate this issue, we integrate MVDream (Shi et al., 2024) with our framework, which is able to generate 3D-aware four view images using input text and camera poses.

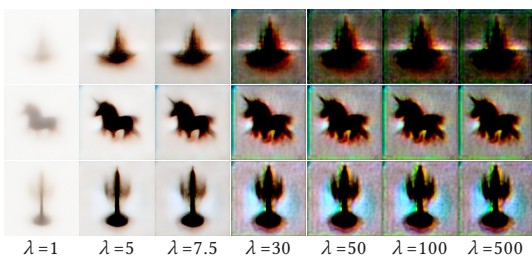

Figure 4: Pseudo ground truth $\hat{I}_0$ visualization. As the CFG weight $\lambda$ decreases, the effective supervision region of $\hat{I}_0$ also decreases.

The impact of the CFG weight $\lambda$ on the optimization process has received attention in existing text-to-3D methods. By visualizing the pseudo ground truth $\hat{I}_0$ after denoising with different levels of weight $\lambda$ at large noise timesteps in Fig. 4, we find that as $\lambda$ decreases, the effective supervision region of $\hat{I}_0$ decreases, hereby impacting the final shape of results.

## 5.2 IMAGE-TO-3D SKETCH

Zero-1-to-3 (Liu et al., 2023) proposes an image-guided SDS method that conditions on the input view image $\tilde{I}$ and relative camera extrinsic $(R, T)$, based on its view-conditioned model. Our framework is able to generate 3D sketch $\tilde{\Theta}$ from the input reference image $\tilde{I}$ by distilling the 3D consistent prior:

$$\nabla_{\tilde{\Theta}} \mathcal{L}_{SDS}(\phi, I = \mathcal{R}(\tilde{\Theta})) = \mathbb{E}_{t,\epsilon} \left[ \omega(t)(\hat{\epsilon}_\phi(I_t; \tilde{I}, R, T, t) - \epsilon) \frac{\partial I}{\partial \tilde{\Theta}} \right]. \quad (13)$$

## 5.3 DYNAMIC NOISE DELETION

We find that our framework is affected by the issue of gradient sparsity in the *Monte Carlo Sampling* strategy. In Reynolds' formula, the gradient from pixel space is determined by the curve boundaries. This results in the loss of information from pixels that are not incident to those boundaries. In cases where significant displacements of curves are required, the optimization tends to get trapped in a local optimum state: curves contract the boundaries to minimize the loss function. During our training process, some curves tend to contract to extremely small lengths as noises. To obtain clean results, we propose the *Dynamic Noise Deletion* strategy. We discretize each curve into $n$ line segments ($n = 20$) and accumulate their lengths as the total length of the curve. The curve with lengths below a specified threshold will be considered as noise and removed dynamically.

## 5.4 TIME ANNEALING SCHEDULE

To further enhance the performance of the results, we employ a time annealing schedule similar with (Wang et al., 2023b; Zhu et al., 2024; Shi et al., 2024; Huang et al., 2024). During the optimization process, we gradually adjust the maximum and minimum time step for SDS in a linear manner. The large noise timestep focuses on aligning the curves with the semantic content of the input text or image during the initial training phase. Conversely, the small noise timestep employed during the later training phase focuses on further enhancing the details.

## 6 EXPERIMENTS

### 6.1 IMPLEMENTATION DETAILS

We implement our rendering framework in C++/CUDA with a PyTorch interface (Paszke et al., 2019). In the experiment, a user-specified number of curves will be randomly initialized within a sphere of radius 1.5. We randomly sample the camera position using the radius from 1.8 to 2.0, with the azimuth in the range of -180 to 180 degrees, the elevation in the range of 0 to 30 degrees and the field of view (fov) of 60 degrees. For the pre-trained model, we apply Stable Diffusion 2.1 (sta) and

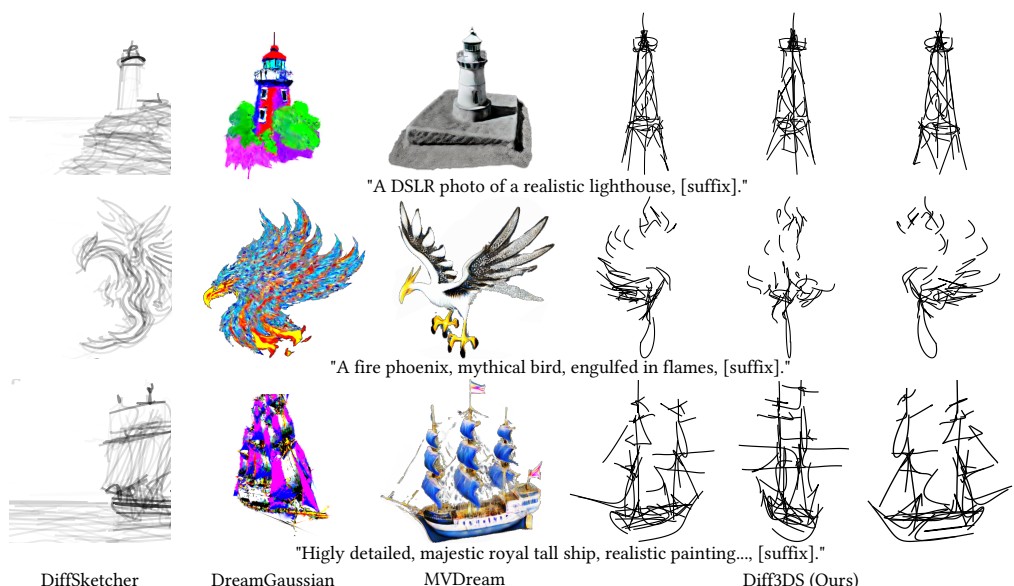

"A DSLR photo of a realistic lighthouse, [suffix]."

"A fire phoenix, mythical bird, engulfed in flames, [suffix]."

"Higly detailed, majestic royal tall ship, realistic painting..., [suffix]."

| DiffSketcher | DreamGaussian | MVDream | Diff3DS (Ours) |

Figure 5: Qualitative results of the text-to-3D sketch task. Existing text-to-3D methods fail to generate sketch-style results, even with the addition of the *"sketch in black and white, line drawing"* suffix.

Table 1: Evaluation of text-to-3D sketch task.

| Method | CLIP-Score$^T$ ↑ | | BLIP-Score ↑ |
|---|---|---|---|
| | ViT B/16 | ViT B/32 | |
| DreamGaussian | 0.2561 | 0.2453 | 0.4990 |
| MVDream | 0.2763 | 0.2653 | 0.4949 |
| Diff3DS (Ours) | **0.3046** | **0.3034** | **0.5038** |

Table 2: Evaluation of image-to-3D sketch task.

| Method | Novel Views | | Reference View |
|---|---|---|---|
| | CLIP-Score$^I$ ↑ | | LPIPS ↓ |
| | ViT B/16 | ViT B/32 | |
| NEF | 0.6612 | 0.6605 | 0.3997 |
| 3Doodle | 0.6734 | 0.6746 | **0.2567** |
| Diff3DS (Ours) | **0.6768** | **0.6846** | 0.2647 |

MVDream (Shi et al., 2024) for the text-to-3D sketch task. And we apply Zero-1-to-3 (Liu et al., 2023) and Stable-Zero123 (Sta, 2023) for the image-to-3D sketch task. The training process requires 1 hour for the text-to-3D sketch task and 2 hours for the image-to-3D sketch task on a single NVIDIA A10 GPU. More details can be found in the Appendix A.

## 6.2 TEXT-TO-3D SKETCH

**Baselines** To the best of our knowledge, we are the first text-to-3D sketch method. We select the existing text-to-3D object methods DreamGaussian (Tang et al., 2024) and MVDream (Shi et al., 2024) as compared baselines, and further choose the text-to-2D sketch method DiffSketcher (Xing et al., 2023) as the additional perceptual reference.

**Qualitative Comparisons** Fig. 5 shows the qualitative results of the text-to-3D sketch task. We compared the 3D results of the text-to-3D baseline method with our method, and further provide the 2D results of DiffSketcher. To encourage baseline methods to generate sketch-style results, we append the suffix *"sketch in black and white, line drawing"* to each prompt inputted to the baselines. From the results, it can be seen, even with the additional suffix, the general text-to-3D baselines DreamGaussian and MVDream cannot generate the sketch-style content. For DiffSketcher, the results are more like sketch draws and only in 2D. In contrast, our method can generate 3D view-consistent sketches with clearly defined curves.

**Quantitative Comparisons** We collect 35 text prompts from previous works (Poole et al., 2023; Shi et al., 2024) and websites. Notably, the accurate evaluation of 3D sketch generation is yet to be resolved due to the absence of ground truth sketches. In this work, we measure the CLIP text-image similarity (Radford et al., 2021) (CLIP-Score$^T$) and BLIP-Score (Li et al., 2022) metrics to evaluate the consistency of the rendered views with the input text prompt, following previous work (Xing et al., 2024). For all metrics, we render the 3D sketch into 8 views and compute the metric between each view and the input text prompt, and use the averaged value as the final result. Table 1 shows the evaluation results and our method outperforms all the text-to-3D baselines.

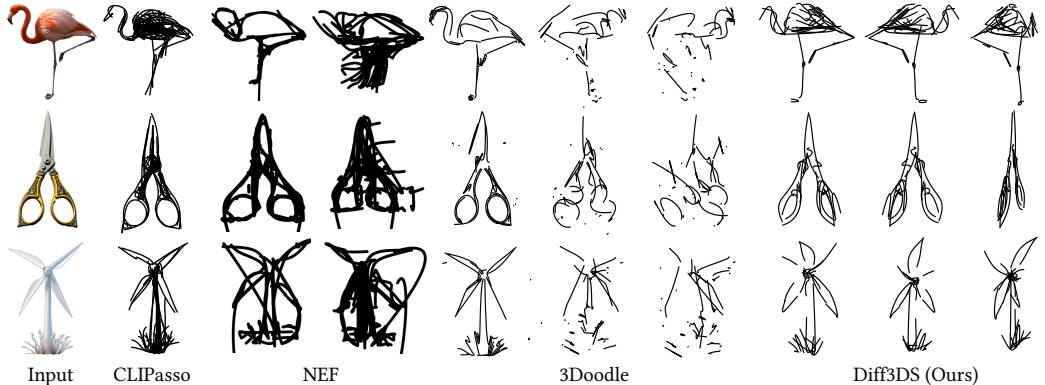

Input    CLIPasso        NEF              3Doodle          Diff3DS (Ours)

Figure 6: Qualitative results of the image-to-3D sketch task. 3Doodle and NEF fail to extract reasonable 3D curves from a single input image.

## 6.3 IMAGE-TO-3D SKETCH

**Baselines**  To the best of our knowledge, we are the first to generate 3D sketch from a single image. The closest works to ours are Neural Edge Fields (NEF) (Ye et al., 2023) and 3Doodle (Choi et al., 2024), which generate 3D curves from multiview images. We also choose the image-to-2D sketch method CLIPasso (Vinker et al., 2022) as the additional perceptual reference.

**Qualitative Comparisons**  Fig. 6 shows the qualitative results of the image-to-3D sketch task. Note that CLIPasso can only generate 2D sketch and is used as a reference for the results of 2D generation methods. As the reconstruction methods, 3Doodle and NEF attain reasonable results for the reference view, but they fail for other views because they lack the ability to predict the unobserved regions from a single input image. In contrast, our method can generate view-consistent 3D sketches.

**Quantitative Comparisons**  We collect 25 images generated with Imagine (Ima, 2023). Follow previous works (Qian et al., 2024; Choi et al., 2024), we use CLIP visual similarity (Radford et al., 2021) (CLIP-Score[I]) metric to measure the abstract semantic similarity across the reference image and the rendered novel views, while employing LPIPS (Zhang et al., 2018) metric in the reference view to measure structural semantic similarity. For the CLIP-Score[I], we calculate the metric between each of the 8 rendered images and the reference image, and use the average value as the final score. Table 2 reports the evaluation results. On the LPIPS metric, our method significantly outperforms NEF and achieves a score comparable to 3Doodle, which has optimized for the reference view with more losses, including the LPIPS and CLIP loss. On the CLIP-Score metric, our method surpasses all baseline approaches. These results demonstrate the superiority of our method.

**User Study**  We further conduct a user study to evaluate the overall quality of our image-to-3D sketch results. Specifically, we prepared 25 tasks, each of which is composed of three randomly-ordered 3D sketches generated using three methods: NEF, 3Doodle and Diff3DS. We used the Likert scores as the evaluation metric, with a range from 1 to 5 where a higher score indicates that the generated 3D sketch is more preferred by the users. In each task, the participants were asked to score each 3D sketch result based on the following two questions: (1) How well does the 3D sketch fit the input image, e.g., in terms of keeping the similar shape and structure of the input? (2) How is the quality of the 3D sketch, e.g., whether the 4 rendered views are 3D consistent or whether the sketch is visually pleasing? We distributed questionnaires to 40 participants who are CS, EE and Math students and researchers, and got 1000 valid scores in total. The results are summarized in Fig. 7, which highlights the superiority of our method. More details can be found in the Appendix D.

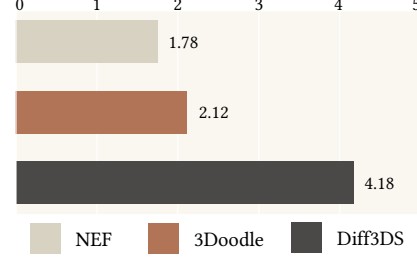

Figure 7: Bar plots of our user study results. The score is of scale 1-5, the higher the better.

### 6.4 ABLATIONS AND ANALYSIS

**Effect of Hyperparameters**   Fig. 8 illustrates the effect of hyperparameters. Fig. 8 (a) illustrates the effect of different CFG weights $\lambda$ in the text-to-3D sketch task. A larger weight leads to a stronger shape prior, while a smaller weight results in shape degradation. Fig. 8 (b) illustrates the effect of various pre-trained models. The Janus Problem easily arises when using Stable Diffusion, and MVDream partially mitigates this issue (e.g., the bicycle with three wheels in the bird's-eye view). Then Zero-1-to-3 is not sufficiently 3D consistent in certain examples (e.g., the upper part of the scissors is bent in the side view). Fig. 8 (c) illustrates that increasing the initial curve number can enhance the details of results, even though the final curve number is automatically determined by *Dynamic Noise Deletion*. Too few curves may fail to represent a complete 3D object.

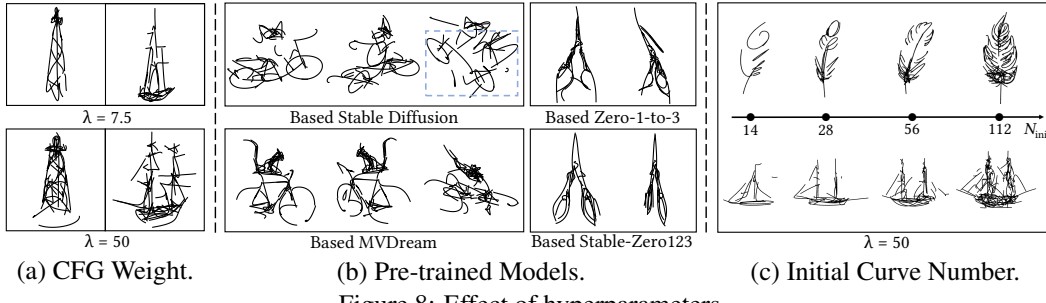

| (a) CFG Weight. | (b) Pre-trained Models. | (c) Initial Curve Number. |

Figure 8: Effect of hyperparameters.

**Analysis of Designed Rasterizer**
Our rasterizer accurately renders the blended colors of overlapping curves and faithfully reproduces occlusion relationships between curves, and enables seamless integration with other 3D sketch generation methods like 3Doodle. To assess its effectiveness, we conduct an experiment to generate colored 3D sketches from multi-view

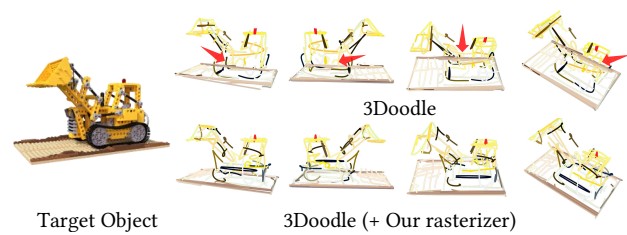

Figure 9: Analysis of designed rasterizer.

images using both the original 3Doodle and a variant integrated with our rasterizer. As shown in Fig. 9, the original 3Doodle suffers from noticeable color conflict errors across multiple viewpoints (e.g., the support base, which is farther from the camera, is incorrectly rendered in the foreground, occluding the LEGO excavator). These noticeable color conflicts disrupt the object's visual coherence and lead to semantic ambiguity. In contrast, our rasterizer consistently reproduces accurate occlusion relationships between objects, providing clearer visual and spatial semantics. Additional results and details can be found in the Appendix D.8.

## 7 CONCLUSIONS

In this paper, we propose Diff3DS, a novel differentiable rendering framework for generating view-consistent 3D sketch by optimizing 3D parametric curves under various supervisions. By employing the recent Score Distillation Sampling (SDS) to distill prior knowledge from pre-trained 2D image generation model, we achieve 3D sketch generation from the flexible text or single image input. Our proposed rasterizer accurately renders the blended colors in overlapping regions and faithfully reproduces occlusion relationships between curves, showcasing its strong potential for generating colored 3D sketches. One limitation of Diff3DS is it inherits the sparse gradient issue from DiffVG and can only optimize continuous parameters. Also, the initial curve number is set manually to achieve a balance between the approximation accuracy and complexity of the results. Moreover, our current implementation does not distinguish between the view-independent curve (e.g., feature lines) and the view-dependent curve (e.g., contours of smooth surface boundaries), limiting the expressive capacity of the overall shape of a 3D object, which is extensively discussed in 3Doodle. Incorporating the diverse curve representations of 3Doodle to Diff3DS is a promising future work. Also, one future direction is to extend our framework for scene-level generation task, such as text-to-3D scene sketch.

## ACKNOWLEDGEMENT

This work was supported in part by National Natural Science Foundation of China (62202199).

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

SUPPLEMENTARY

OVERVIEW

This supplementary material is structured into multiple sections that offer further details and analysis pertaining to our Diff3DS research. Concretely, it will hone in on the topics that follow:

- In section A, we provide the implementation details of Diff3DS.
- In section B, we provide a detailed proof for the 3D rational Bézier curve projection.
- In section C, we provide a detailed analysis for finding the closest distance between a point and a rational Bézier curve.
- In section D, we provide more analysis and experiment results.
- In section E, we provide more high quality sketch results.

## A  IMPLEMENTATION DETAILS

We implement our rendering framework in C++/CUDA with the PyTorch interface, and follow the STEP standard (ISO 10303-21) (Loffredo, 1999) to format and save our 3D curve results. In the experiment, a user-specified number of curves will be randomly initialized within a sphere of radius 1.5. The default curve number is set to 56. We randomly sample the camera position using the radius from 1.8 to 2.0, with the azimuth in the range of -180 to 180 degrees, the elevation in the range of 0 to 30 degrees and the field of view (fov) of 60 degrees. Notice that Stable-Zero123 abnegates the relative distance for simplifying [2], we fix its radius at 2.0.

For the pre-trained model, we apply Stable Diffusion 2.1 (sta) and MVDream (Shi et al., 2024) for the text-to-3D sketch task, with the CFG weight of 50. And we apply Zero-1-to-3 (Liu et al., 2023) and Stable-Zero123 (Sta, 2023) for the image-to-3D sketch task, with the CFG weight of 7.5 followed by (Liu et al., 2023; Sta, 2023) .

For all tasks, the total number of training steps is 4000. Starting from step 2000, we dynamically delete the noise every 100 steps. The training process requires 1 hour for the text-to-3D sketch task and 2 hours for the image-to-3D sketch task on a single NVIDIA A10 GPU with a batch size of 4. To optimize the control point positions, we use the Adam optimizer and set the learning rate of the optimizer to 0.002. For the time annealing schedule, we prefer to decrease the maximum and minimum time steps from 0.85 to 0.3 and 0.1, respectively, over the first 3600 steps. Due to limited resources, not all possible combinations of the hyper-parameters related to the time steps have been fully explored, and there may be other configurations that could produce better results.

For the *Dynamic Noise Deletion*, only the curves with lengths below a specified threshold will be considered as noise and removed dynamically. The trade-off between noise removal and detail preservation can be balanced by adjusting the threshold. Currently, the threshold is empirically set to 0.1 which is below the $10\%$ of the average curve length.

## B  3D RATIONAL BÉZIER CURVE PROJECTION

Given the control points $\tilde{P}_i = (\tilde{P}_i^x, \tilde{P}_i^y, \tilde{P}_i^z) \in \mathbb{R}^3$, the associated non-negative weights $\tilde{w}_i$ and the Bernstein polynomials $B_{i,n}(t)$, for $0 \leq t \leq 1$, define a $n$th-degree 3D rational Bézier curve $\tilde{p}(t)$ by:

$$\tilde{p}(t) = \sum_{i=0}^{n} \frac{B_{i,n}(t)\tilde{w}_i}{\sum_{j=0}^{n} B_{j,n}(t)\tilde{w}_j} \tilde{P}_i = (\tilde{x}(t), \tilde{y}(t), \tilde{z}(t)).$$

Assume the focal plane is $z = f$, where $f$ is the focal length, the corresponding projected 2D curve $p(t)$ under the pinhole camera perspective projection can be defined as:

$$p(t) = (x(t), y(t)) = (\frac{\tilde{x}(t)}{\tilde{z}(t)}f, \frac{\tilde{y}(t)}{\tilde{z}(t)}f).$$

---

[2]https://github.com/threestudio-project/threestudio/issues/360

Due to rational Bézier curves are projective invariant, the projected 2D curve $p(t)$ still is identical to the rational Bézier curve defined by projected control points. Given projected 2D control points

$$P_i = (P_i^x, P_i^y) = (\frac{\tilde{P}_i^x}{\tilde{P}_i^z}f, \frac{\tilde{P}_i^y}{\tilde{P}_i^z}f) \in \mathbb{R}^2,$$

$p(t)$ can be written as a 2D rational Bézier curve with adjusted weights $w_i = \tilde{w}_i \tilde{P}_i^z$:

$$
\begin{aligned}
p(t) &= (\frac{\tilde{x}(t)}{\tilde{z}(t)}f, \frac{\tilde{y}(t)}{\tilde{z}(t)}f) \\
&= (\frac{\sum_{i=0}^n B_{i,n}(t)\tilde{w}_i \tilde{P}_i^x}{\sum_{j=0}^n B_{j,n}(t)\tilde{w}_j \tilde{P}_j^z}f, \frac{\sum_{i=0}^n B_{i,n}(t)\tilde{w}_i \tilde{P}_i^y}{\sum_{j=0}^n B_{j,n}(t)\tilde{w}_j \tilde{P}_j^z}f) \\
&= (\sum_{i=0}^n \frac{B_{i,n}(t)(\tilde{w}_i \tilde{P}_i^z)}{\sum_{j=0}^n B_{j,n}(t)(\tilde{w}_j \tilde{P}_j^z)} \left(\frac{\tilde{P}_i^x}{\tilde{P}_i^z}f\right), \sum_{i=0}^n \frac{B_{i,n}(t)(\tilde{w}_i \tilde{P}_i^z)}{\sum_{j=0}^n B_{j,n}(t)(\tilde{w}_j \tilde{P}_j^z)} \left(\frac{\tilde{P}_i^y}{\tilde{P}_i^z}f\right)) \\
&= (\sum_{i=0}^n \frac{B_{i,n}(t)w_i}{\sum_{j=0}^n B_{j,n}(t)w_j} P_i^x, \sum_{i=0}^n \frac{B_{i,n}(t)w_i}{\sum_{j=0}^n B_{j,n}(t)w_j} P_i^y) \\
&= (x(t), y(t)) \\
&= \sum_{i=0}^n \frac{B_{i,n}(t)w_i}{\sum_{j=0}^n B_{j,n}(t)w_j} P_i.
\end{aligned}
$$

## C  CLOSEST DISTANCE BETWEEN A POINT AND A RATIONAL BÉZIER CURVE

Given a point $q \in \mathbb{R}^2$ and a $n$th-degree rational Bézier curve

$$p(t) = \frac{\sum_{i=0}^n B_{i,n}(t)w_i P_i}{\sum_{j=0}^n B_{j,n}(t)w_j},$$

we want to find the closest distance between them and the closest point $t^\star$ on the curve $p$. The closest distance is defined as $(p(t^\star) - q)^2$, and $t^\star$ is defined as:

$$t^\star = \arg_t \min(p(t) - q)^2.$$

To solve the $t^\star$, we take the derivative of the squared distance with respect to $t$ and set it to zero:

$$2(p(t) - q)p'(t) = \rho(t) = 0,$$

which equal to solve the roots of a polynomial with degree $3n - 2$. For the quadratic rational curve and the cubic rational curve, a 4th order polynomial and a 7th order polynomial need to be solved respectively. We provide the detail separately in section C.1 and section C.2.

To solve all real roots of these polynomials, inspired by DiffVG, we solve the polynomial using bisection and the Newton-Raphson method (William H. Press & Flannery., 2007). The Newton-Raphson solver obtains its initial guess for intervals from isolator polynomials (Sederberg & Chang., 1994). For the polynomial $\rho(t)$ with two adjacent real roots $t_1$ and $t_2$, given any two lower order other polynomials $b(t)$ and $c(t)$, define

$$a(t) = b(t)\rho'(t) + c(t)\rho(t),$$

where $\rho'(t)$ is the derivative, (Sederberg & Chang., 1994) proofs that $a(t)$ or $b(t)$ must has at least one real root in the closed interval $[t_0, t_1]$. Since $\rho(t_0) = \rho(t_1) = 0$, we know $a(t_0)a(t_1) = b(t_0)b(t_1)\rho'(t_0)\rho'(t_1)$. Since $t_0$ and $t_1$ are roots of $\rho$ and $\rho'(t_0)\rho'(t_1) \leq 0$. Thus either $a(t_0)a(t_1) \leq 0$ or $b(t_0)b(t_1) \leq 0$. Please see the original paper for discussions on multiple roots.

After solving all the real roots of $a$ and $b$ within the $(0, 1)$, the intervals for finding the real roots of $\rho$ are then determined. We can then use bisection and the Newton-Raphson method to find all the roots of $\rho$ within these intervals. The key operation is the selection of the polynomials $b(t)$ and $c(t)$, (Sederberg & Chang., 1994) prefer to find the $b(t)$ and $c(t)$ which follow $\text{Degree}(a) + \text{Degree}(b) = \text{Degree}(\rho) - 1$. In the upcoming paragraphs, we will discuss the cases of the 4th order polynomial and the 7th order polynomial in section C.3 and section C.4.

### C.1 CLOSEST DISTANCE OF QUADRATIC RATIONAL CURVE CASE

The quadratic rational curve $p(t)$ can be reorganized using variables $A$ through $F$ as follows:

$$p(t) = \frac{(1-t)^2 w_0 P_0 + 2(1-t)tw_1 P_1 + t^2 w_2 P_2}{(1-t)^2 w_0 + 2(1-t)tw_1 + t^2 w_2} = \frac{At^2 + Bt + C}{Dt^2 + Et + F},$$

we can further calculate $p(t) - q$ and $p'(t)$ as:

$$p(t) - q = \frac{(A-Dq)t^2 + (B-Eq)t + (C-Fq)}{Dt^2 + Et + F}$$

$$p'(t) = \frac{(AE-BD)t^2 + (2AF-2CD)t + (BF-CE)}{(Dt^2 + Et + F)^2}$$

By reorganizing them using variables $a$ through $f$:

$$p(t) - q = \frac{at^2 + bt + c}{Dt^2 + Et + F}$$

$$p'(t) = \frac{dt^2 + et + f}{(Dt^2 + Et + F)^2},$$

we can rewrite the target polynomial $\rho(t)$ as below:

$$\rho(t) = 2(p(t) - q)p'(t) = 2\frac{(at^2 + bt + c)(dt^2 + et + f)}{(Dt^2 + Et + F)^3},$$

whose numerator is a 4th order polynomial in the variable $t$. Note that the denominator of $\rho(t)$ is strictly positive, due to the definition of the rational Bézier curve. Therefore, finding the real roots of the equation $\rho(t) = 0$ is equivalent to finding the real roots of the 4th order polynomial in the numerator.

### C.2 CLOSEST DISTANCE OF CUBIC RATIONAL CURVE CASE

The cubic rational curve $p(t)$ can be reorganized using variables $A$ through $H$ as follows:

$$p(t) = \frac{(1-t)^3 w_0 P_0 + 3(1-t)^2 tw_1 P_1 + 3(1-t)t^2 w_2 P_2 + t^3 w_3 P_3}{(1-t)^3 w_0 + 3(1-t)^2 tw_1 + 3(1-t)t^2 w_2 + t^3 w_3}$$

$$= \frac{At^3 + Bt^2 + Ct + D}{Et^3 + Ft^2 + Gt + H},$$

we can further calculate $p(t) - q$ and $p'(t)$ as:

$$p(t) - q = \frac{(A-Eq)t^3 + (B-Fq)t^2 + (C-Gq)t + (D-Hq)}{Et^3 + Ft^2 + Gt + H}$$

$$p'(t) = \frac{3A^2 + 2Bt + C}{Et^3 + Ft^2 + Gt + H} - \frac{(3Et^2 + 2Ft + G)(At^3 + Bt^2 + Ct + D)}{(Et^3 + Ft^2 + Gt + H)^2}$$

By reorganizing them using variables $a$ through $i$:

$$p(t) - q = \frac{at^3 + bt^2 + ct + d}{Et^3 + Ft^2 + Gt + H}$$

$$p'(t) = \frac{et^4 + ft^3 + gt^2 + ht + i}{(Et^3 + Ft^2 + Gt + H)^2},$$

we can write the target polynomial $\rho(t)$ as below:

$$\rho(t) = 2(p(t) - q)p'(t) = 2\frac{(at^3 + bt^2 + ct + d)(et^4 + ft^3 + gt^2 + ht + i)}{(Et^3 + Ft^2 + Gt + H)^3},$$

whose numerator is a 7th order polynomial in the variable $t$. Note that the denominator of $\rho(t)$ is strictly positive, due to the definition of the rational Bézier curve. Therefore, finding the real roots of the equation $\rho(t) = 0$ is equivalent to finding the real roots of the 7th order polynomial in the numerator.

### C.3 ISOLATOR POLYNOMIALS OF THE 4TH ORDER POLYNOMIAL CASE

We aim to isolate the real roots of the 4th order polynomial $\rho(t)$ by the roots of two lower order polynomials $a(t)$ and $b(t)$. To keep $\text{Degree}(a) + \text{Degree}(b) = \text{Degree}(\rho) - 1$, we select $a(t)$ as the 2nd order polynomial and $b(t)$ as the linear polynomial.

Define $\rho(t)$ and $\rho'(t)$ as follows:

$$\rho(t) = t^4 + Bt^3 + Ct^2 + Dt + E$$
$$\rho'(t) = 4t^3 + 3Bt^2 + 2Ct + D,$$

and we know $a(t) = c(t)\rho(t) - b(t)\rho'(t)$. Let $c(t) = 1$, $a(t)$ can be written as $a(t) = \rho(t) - b(t)\rho'(t)$ By doing a long division between $\rho$ and $\rho'$, we obtain the isolator polynomials as follows:

$$a(t) = (\frac{C}{2} - \frac{3B^2}{16})t^2 + (\frac{3D}{4} - \frac{CB}{8})t + (e - \frac{DB}{16})$$
$$b(t) = \frac{1}{4}t + \frac{B}{16}$$

### C.4 ISOLATOR POLYNOMIALS OF THE 7TH ORDER POLYNOMIAL CASE

We aim to isolate the real roots of the 7th order polynomial $\rho(t)$ by the roots of two lower order polynomials $a(t)$ and $b(t)$. To keep $\text{Degree}(a) + \text{Degree}(b) = \text{Degree}(\rho) - 1$, we select $a(t)$ and $b(t)$ both be the cubic polynomial.

Define $\rho(t)$ and $\rho'(t)$ as follows:

$$\rho(t) = t^7 + p_6 t^6 + .... + p_1 t + p_0$$
$$\rho'(t) = 7t^6 + 6p_6 t^5 + .... + p_1,$$

and we know $a(t) = c(t)\rho(t) - b(t)\rho'(t)$. We wish $a$ and $b$ to both be cubic polynomials. Note that $b(t)p'(t)$ is a 9th order polynomial, so we assume $c(t)$ to be a 2nd order polynomial to ensure that $c(t)p(t)$ is a 9th order polynomial too. Let $b(t) = (t^3 + Ct^2 + Dt + E)$ and $c(t) = (7t^2 + At + B)$, we can obtain two 9th order polynomials, and their difference a(x) is a cubic polynomial:

$$c(t)\rho(t) = (t^7 + p_6 t^6 + .... + p_1 t + p_0)(7t^2 + At + B)$$
$$b(t)\rho'(t) = (7t^6 + 6p_6 t^5 + .... + p_1)(t^3 + Ct^2 + Dt + E).$$

By setting the coefficients of the 8th to 4th order terms of $a(t)$ to 0, we can obtain a system of 5 linear equations in variables A through E. By solving this system of equations, we can determine the values of variables A through E. We can then proceed to find $b(t)$, $c(t)$, and $a(t)$. Please refer to our code for the specific details.

## D MORE ANALYSIS AND EXPERIMENTS

### D.1 ABLATION STUDY OF DESIGNED COMPONENTS

We evaluate the contributions of the *Time Annealing Schedule* and *Dynamic Noise Deletion* components, as shown in Fig. 10. The omission of *Time Annealing Schedule* affects the conformity of the generated results with the semantic meaning of the input text or the level of detail in the reference image, and the omission of *Dynamic Noise Deletion* leads to the retention of some noise in the results.

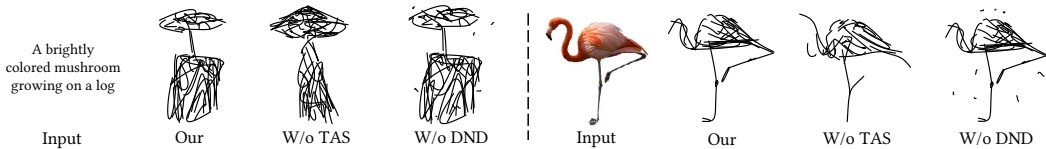

Figure 10: Ablation study on *Time Annealing Schedule* (TAS) and *Dynamic Noise Deletion* (DND).

## D.2 USER STUDY

We provide details of our user study. Specifically, our user study includes 25 tasks, which cover a diverse range of object types, e.g., flamingo, unicorn, windmill, lighthouse, boat, bike, sword, bow, scissor, flower etc. Each task is composed of three 3D sketches generated using three methods: NEF, 3Doodle and Diff3DS. We used the Likert scores as the evaluation metric, with a range from 1 to 5 where a higher score indicates that the generated 3D sketch is more preferred by the users.

In each task, the order of sketches presented to the users was randomized. Specifically, for each method, the 3D sketch is rendered in four views and organized in the form shown in Fig. 11. Then, for each task, it contains three randomly-ordered rows, while each row represents the results of one method. The participants were asked to score each 3D sketch output based on the following two questions:

1. How well does the 3D sketch fit the input image, e.g., in terms of keeping the similar shape and structure of the input?

2. How is the quality of the 3D sketch, e.g., whether the 4 rendered views are 3D consistent or whether the sketch is visually pleasing?

We distributed questionnaires to 40 participants who are CS, EE and Math students and researchers, and finally collected total 1000 valid scores.

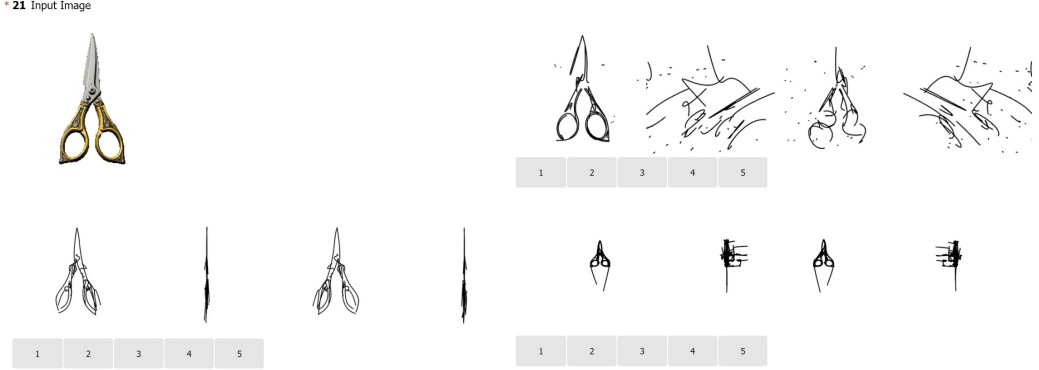

Figure 11: Screenshot of an example question used in our user study

## D.3 EFFICIENCY OF THE INITIAL NUMBER OF CURVES

The effect of the initial curve number have reported in Fig. 8 (c). To further evaluate the efficiency of the curve number, we have conducted a new ablation study on the initial curve number, and the results are shown in Table 3. The results show that as the number of initial curve increases, CLIP-Score performance shows a consistent increase while the average optimization time keeps getting longer. Meanwhile, when the initial number reaches 56, further increasing it to 112 does not result in a significant performance improvement due to the *Dynamic Noise Deletion* process.

| Metric \ Number | num=112 | num=56 | num=28 | num=14 |
|---|---|---|---|---|
| CLIP-Score$^{\mathrm{T}}$ (ViT B/16) | 0.3074 | 0.3046 | 0.2920 | 0.2785 |
| CLIP-Score$^{\mathrm{T}}$ (ViT B/32) | 0.3032 | 0.3034 | 0.2883 | 0.2786 |
| Average Training Time | 80min | 60min | 45min | 35min |

Table 3: Efficiency of the initial curve number.

## D.4 EFFECT OF NOISE SAMPLE SCHEDULE

The choice of noise sampling schedule has a significant impact on the results. For our task, the large noise timestep focuses on aligning the curves with the semantic content of the input condition during the initial training phase. Conversely, the small noise timestep employed during the later training phase focuses on further enhancing the details. In the current implementation, we have employed a biased sampling that decreases the maximum and minimum time steps from 0.85 to 0.3 and 0.1, whereas original DreamFusion randomly samples noises from 0.98 to 0.02. Based on our experience, further reducing the sampling of high-level noise would lead to a significant decline in the geometric quality of the results. We have provided relevant results in the Fig. 12.

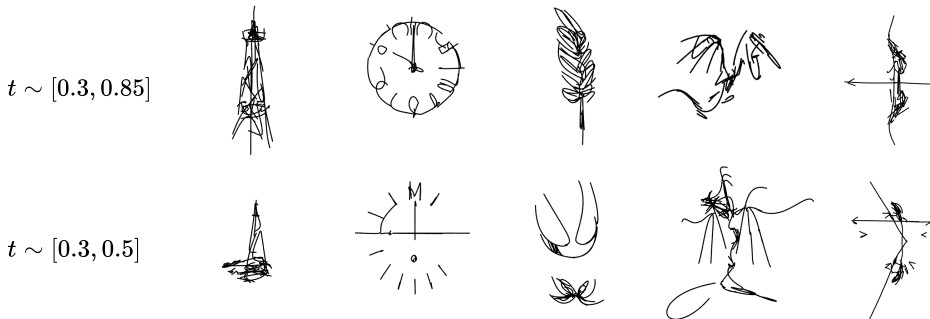

Figure 12: Effect of noise sample schedule.

## D.5 EFFECT OF CURVE INITIALIZATION

In the current implementation, the initial curves are randomly initialized within a sphere of radius 1.5. Based on our observations, the quality of the results is generally insensitive to minor variations in the curve initialization radius. However, significantly reducing the radius can still degrade the quality of the results, as shown in the Fig. 13. We recommend adjusting the radius to ensure the projected curve remains fully visible within the image and occupies a reasonable area.

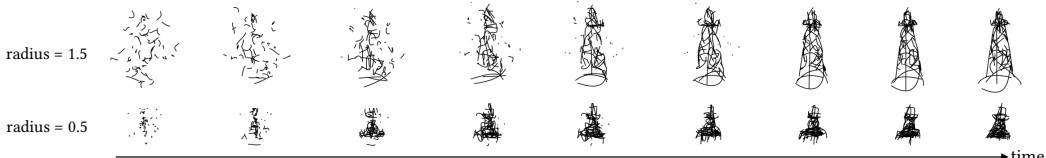

Figure 13: Effect of curve initialization.

## D.6 EFFECT OF OPTIMIZING COLOR, OPACITY AND WIDTH

Our proposed differentiable rasterizer supports the optimization of the control point position, color, opacity and curve width. We report the optimization results in different modes, as shown in Fig. 14, including Position, RGB, RGBA (both color and opacity), Alpha, and Width. Notably, we observe that optimizing the curve width results in unstable outcomes. We hypothesize that this instability stems from the rendered image semantics being highly sensitive to variations in curve width, and plan to explore this topic in our future work.

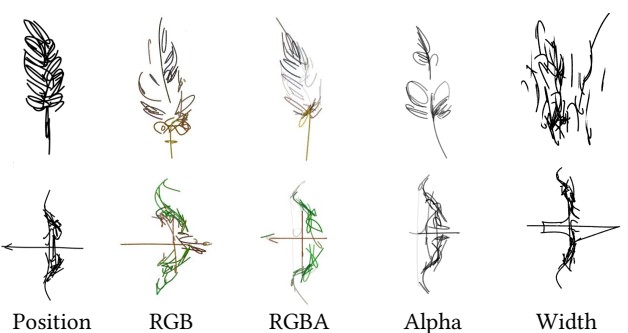

Figure 14: Effect of optimizing color, opacity and width.

### D.7 COMPARISON WITH 3D SKETCH RECONSTRUCTION METHODS UNDER MULTI-VIEW SUPERVISION

We further compare our approach with the 3D sketch reconstruction method under multi-view supervision. Specifically, we employ MVDream and Stable-Zero123 to generate 3D objects from input text and a single view, rendering 120 horizontal views as training data. Notably, we report comparisons only between Diff3DS and 3Doodle, as NEF failed to converge and produce reasonable results in our experiments. To ensure a fair comparison, 3Doodle adopts the same setup as our method, optimizing only the 3D curves and randomly initializing the positions of the curve control points. For the text-to-3D and image-to-3D sketch tasks, 3Doodle requires 3 hours (1 hour for object generation) and 2.5 hours (0.5 hours for object generation), respectively. In contrast, our method only requires 1 hour and 2 hours, respectively.

As illustrated in Fig. 15, our method achieves performance comparable to the baseline within a more efficient training time.

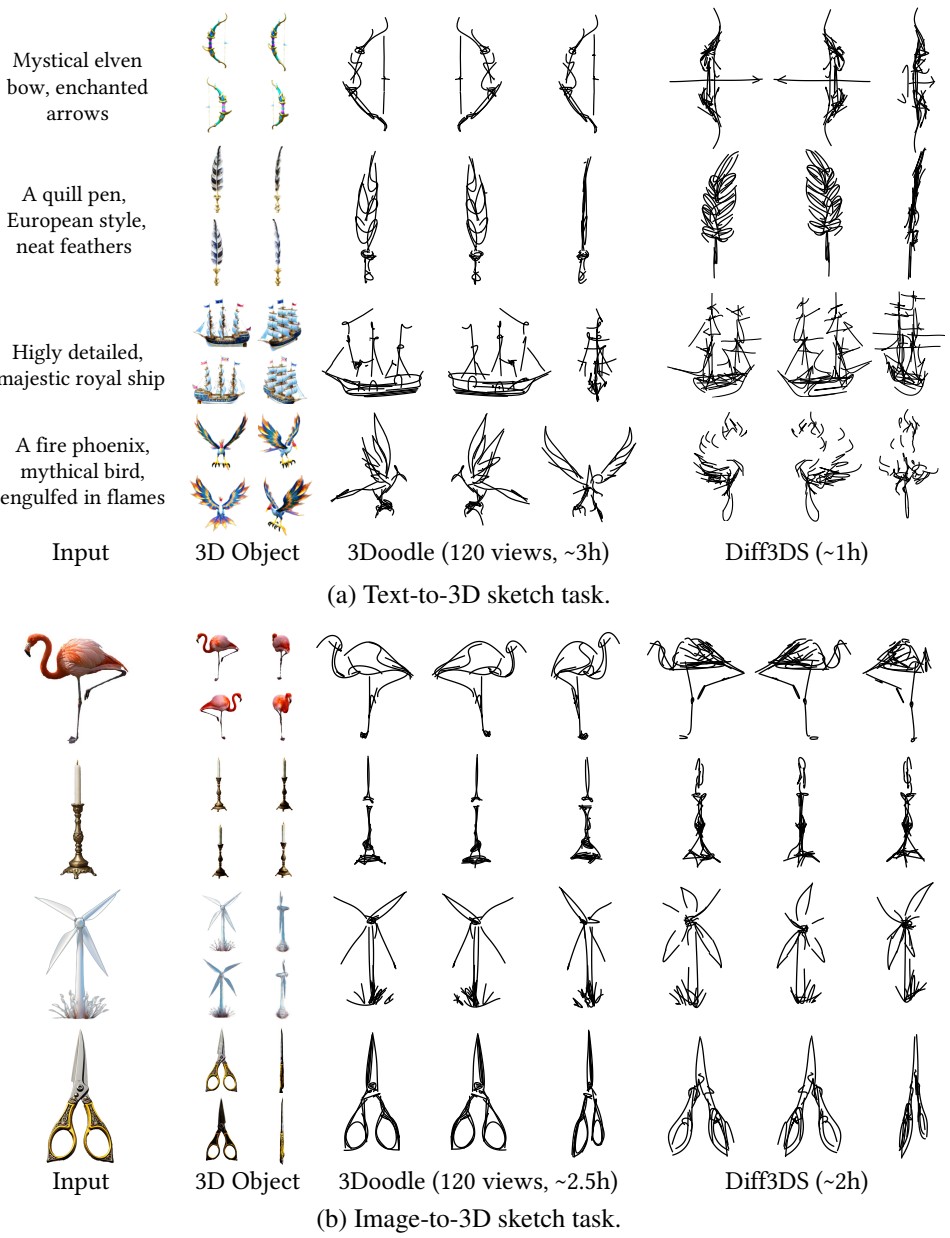

(a) Text-to-3D sketch task.

(b) Image-to-3D sketch task.

Figure 15: Comparison with 3D sketch reconstruction method under multi-view supervision.

## D.8 ANALYSIS OF DESIGNED RASTERIZER

We conduct an experiment generating colored 3D sketches from given multi-view images using both the original 3Doodle and its variant integrated with our rasterizer. For the training dataset, we use the "toyhorse" and "toycar" from the 3Doodle-provided dataset, along with "hotdog", "ship", and "lego" from the Nerf Synthesis dataset (Mildenhall et al., 2021). As shown in Fig. 16, the original 3Doodle suffers from noticeable color conflicts across multiple viewpoints (e.g., the plate's curve incorrectly appears over the hotdog), disrupting the visual coherence of the object and leading to semantic ambiguity. In contrast, our rasterizer consistently reproduces accurate occlusion relationships between objects, providing clearer visual and spatial semantics.

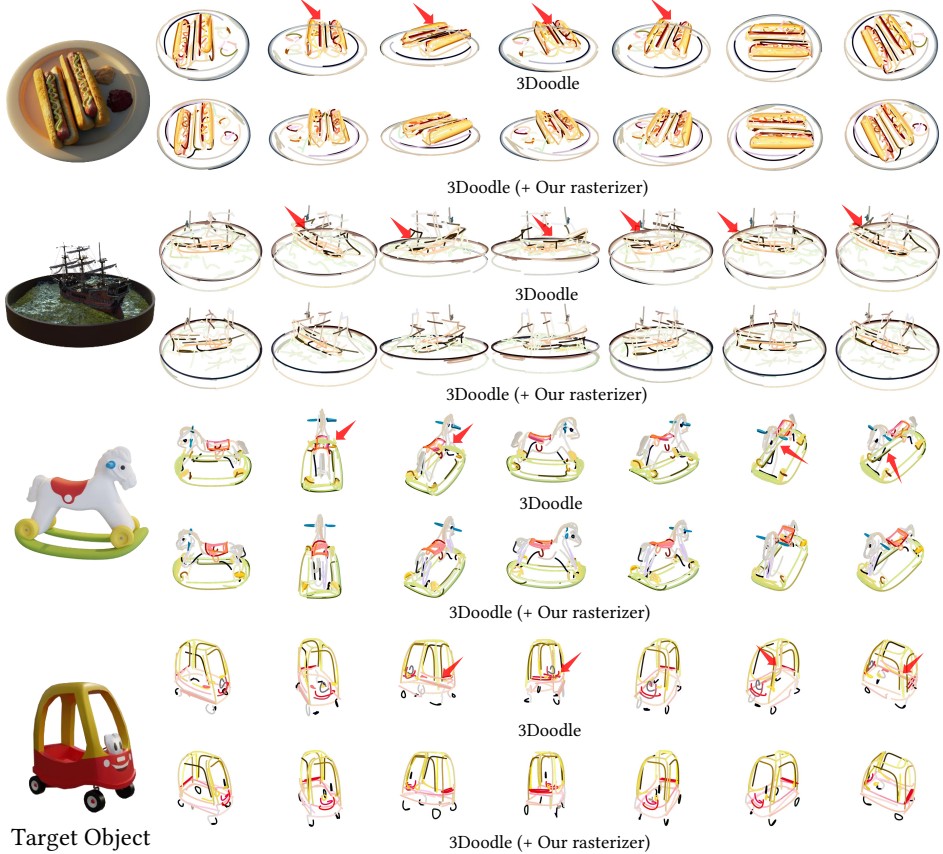

Figure 16: Analysis of designed rasterizer. In the figure, the red arrow points to the curves which has incorrect depth order, e.g., the curve of the plate falsely appears over the hotdog.

## D.9 SKETCH STYLIZATION

The style of our generated sketches can be altered by applying different brushes to the vector strokes. We rendered the 3D sketch in the vector graphics format and used various brush styles from Adobe Illustrator to demonstrate the diverse styles of the generated sketches, as shown in Fig. 17.

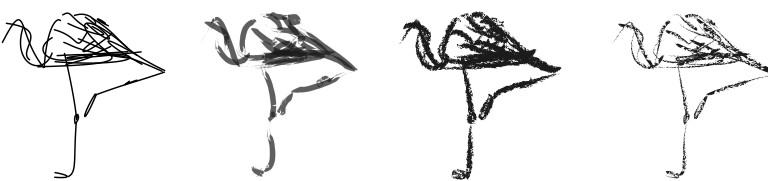

Figure 17: Sketch stylization.

# E    MORE HIGH QUALITY RESULTS

## E.1    MORE RESULTS OF THE TEXT-TO-3D SKETCH TASK

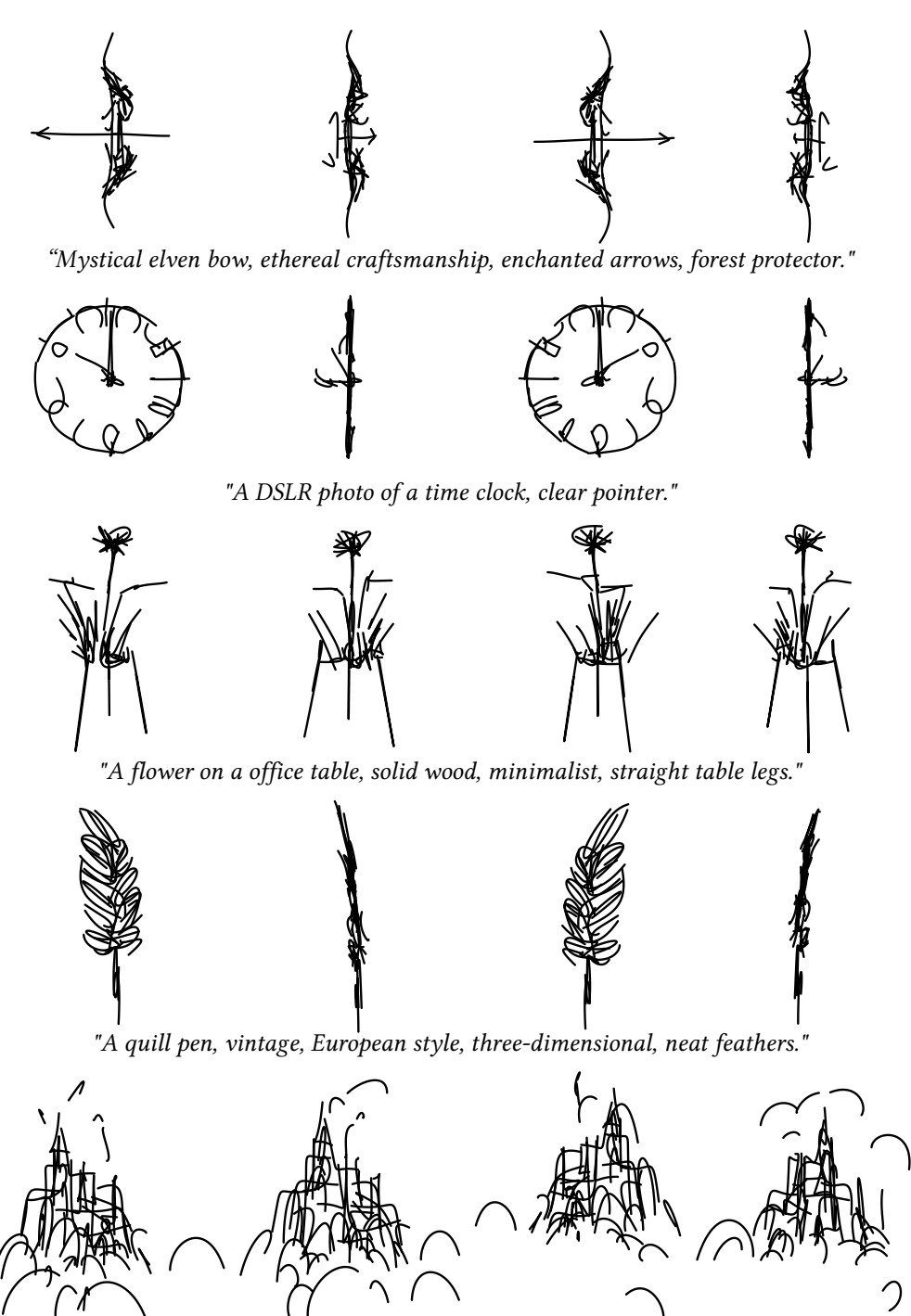

*"Mystical elven bow, ethereal craftsmanship, enchanted arrows, forest protector."*

*"A DSLR photo of a time clock, clear pointer."*

*"A flower on a office table, solid wood, minimalist, straight table legs."*

*"A quill pen, vintage, European style, three-dimensional, neat feathers."*

*"Castle in the clouds, ethereal fortress, sky-high citadel."*

Figure 18: More results of the text-to-3D sketch task.

## E.2 MORE RESULTS OF THE IMAGE-TO-3D SKETCH TASK

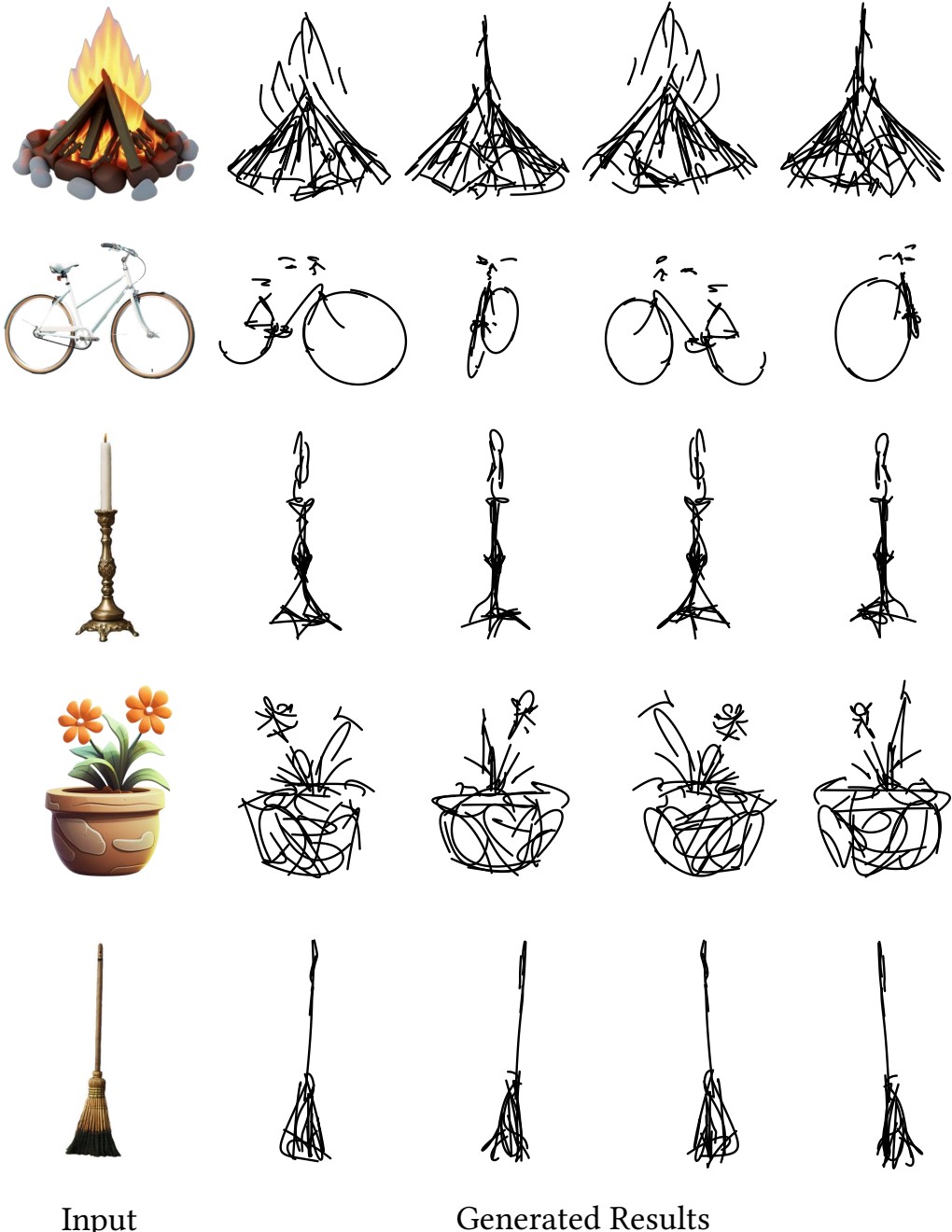

Input          Generated Results

Figure 19: More results of the image-to-3D sketch task

E.3  MORE 3D SKETCH RESULTS RENDERED USING BLENDER COMMUNITY (2018)

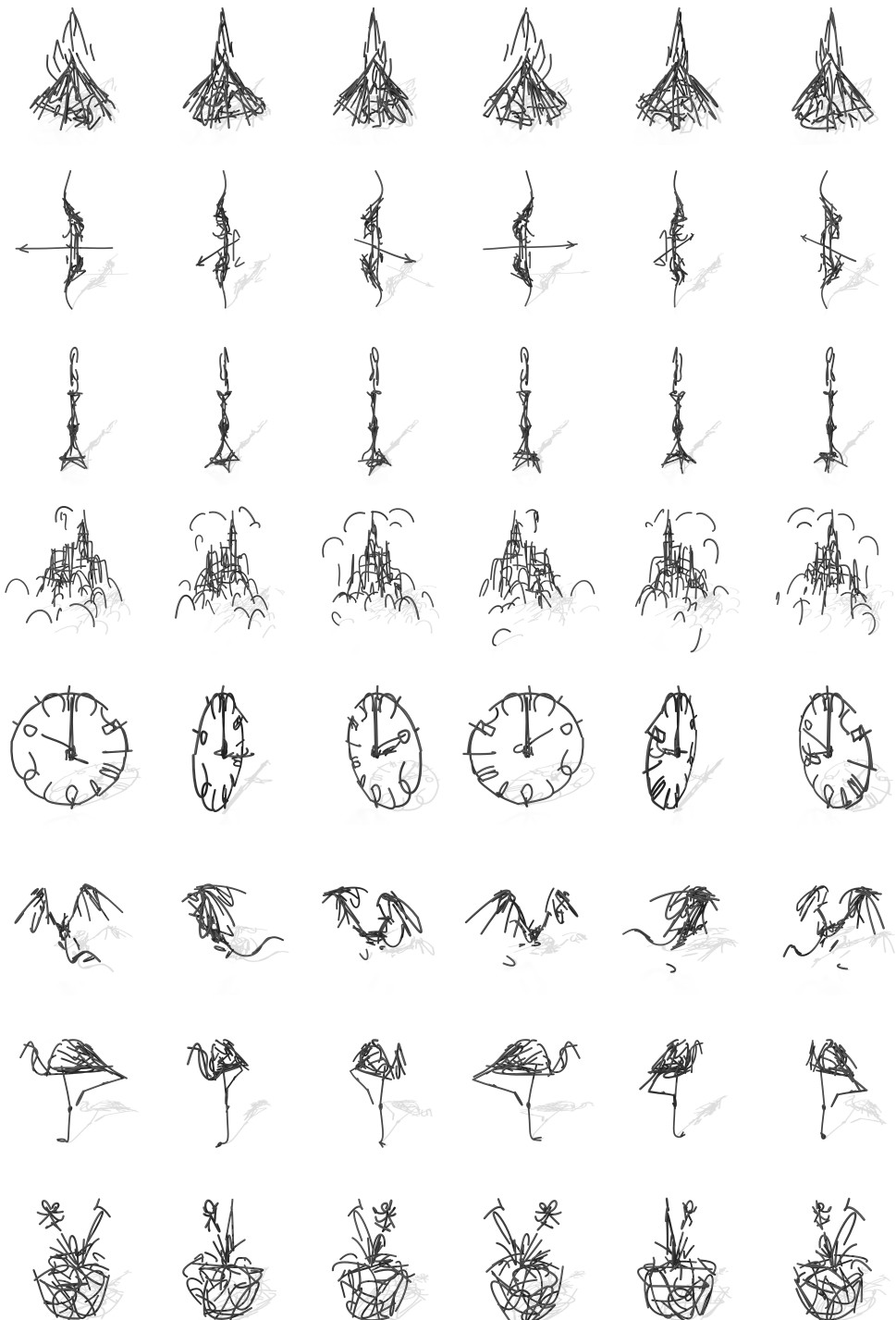

Figure 20: More rendered 3D sketch results.

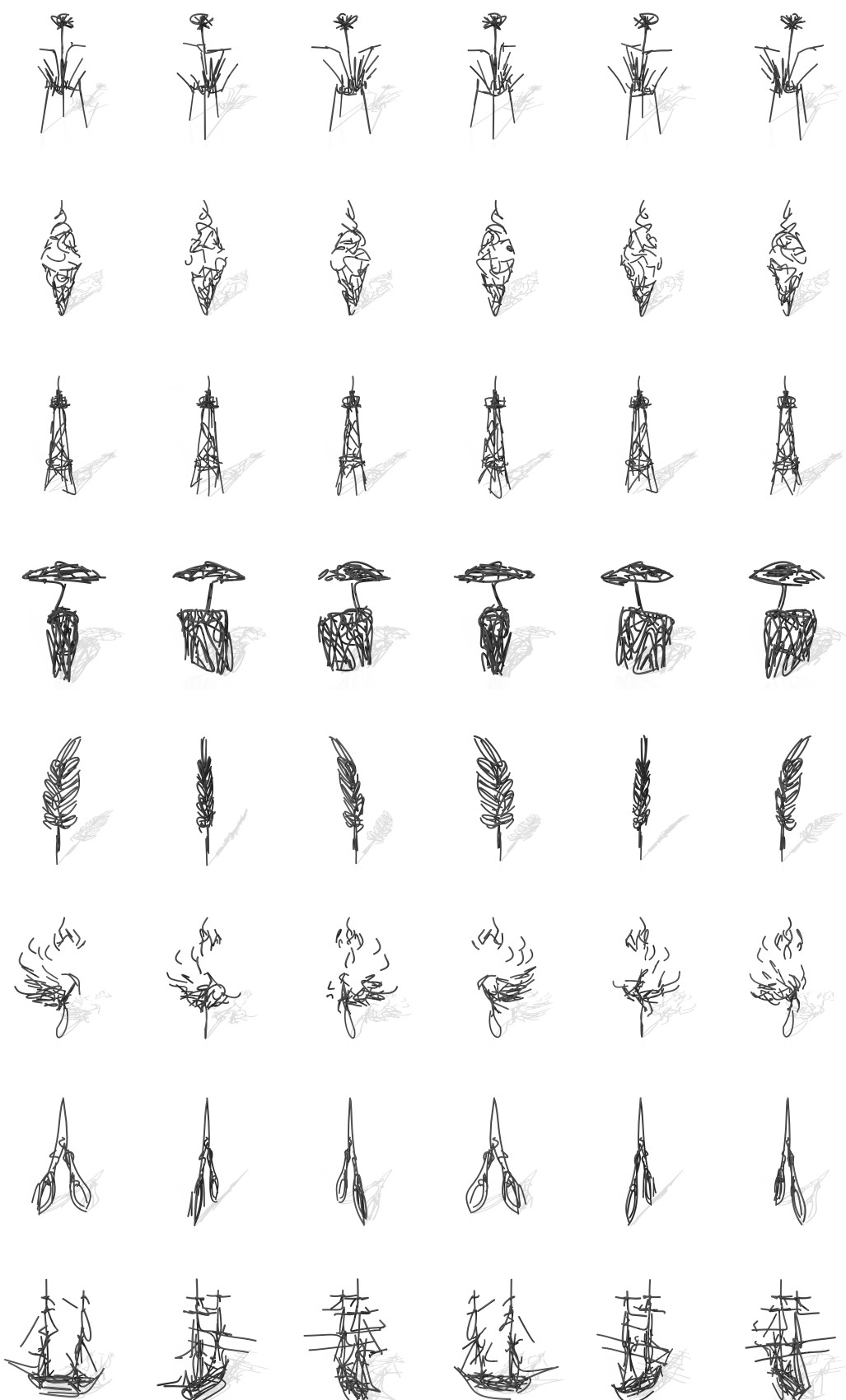

Figure 21: More rendered 3D sketch results.

