# OpenReview forum: "Diff3DS: Generating View-Consistent 3D Sketch via Differentiable Curve Rendering"
_ICLR.cc/2025/Conference — ICLR 2025 Poster_

### Official Review · Reviewer_TMKm · 2024-10-19

**Soundness:** 3
**Presentation:** 3
**Contribution:** 2
**Rating:** 6
**Confidence:** 4

**Summary:**

The paper introduces Diff3DS, a novel approach for generating view-consistent 3D sketches from either text prompts or image inputs. It employs a differentiable rendering framework that enables the optimization of 3D curve parameters through fully differentiable rasterized images. The method first projects parametric 3D curve control points into 2D using perspective projection. It then applies differentiable rasterization, similar to DiffVG, while correctly handling depth during curve intersections. The framework leverages Score Distillation Sampling (SDS) for supervision, using either a pretrained text-to-image model or an image-guided model. Additional techniques, such as dynamic noise deletion and time annealing schedules, further enhance optimization and generation quality.
Experiments show that Diff3DS generates consistent 3D sketches from both text and image inputs, outperforming previous sketch synthesis methods in terms of CLIP visual similarity and user preference.

**Strengths:**

1. The paper introduces a novel framework for 3D sketch generation, which, to the best of my knowledge, is the first to support generation from both images and text prompts.

2. The proposed method employs differentiable rendering that supports perspective projection and occlusion-aware rasterization for 3D curves, distinguishing it from earlier approaches.

3. Experimental results demonstrate that Diff3DS outperforms prior methods in both semantic alignment and user-preferred quality.

4. The writing is clear and the structure of the paper is easy to follow.

**Weaknesses:**

Although the paper is well-written and the generated sketches align with the given image and text inputs, there are concerns about the sufficiency of the contribution and the depth of the experimental evaluation:

1. One of the claimed novelty is the use of perspective projection with depth-aware rasterization. However, it seems this approach is not entirely new. A similar perspective projection has already been discussed in 3Doodle (specifically in Appendix A.2), which seems to have no clear difference with the procedure proposed in this paper.

2. While fixing occlusions in 3D curves is a positive addition to the rasterizer, there is no conclusive experimental evidence showing how this improvement impacts the quality of generation. For instance, it remains unclear whether depth-aware rasterization significantly enhances the final results beyond the occlusion analysis in Figure 11.

3. Another novelty is the use of SDS loss supervision, which has not been employed by previous methods. However, applying SDS loss to rasterized images is relatively straightforward and does not present a significant technical difficulty, given its widespread use in NeRF-related tasks. This loss could be easily incorporated into prior approaches without altering their core components. Moreover, the experiments do not convincingly demonstrate the advantage of direct SDS supervision on 3D curves, which I elaborate below.

4. In the image-to-3D sketch task, Diff3DS is compared with 3Doodle and NEF, showing better CLIP scores and improved 3D consistency. However, the fairness of this comparison is questionable, as Diff3DS benefits from multi-view supervision via SDS loss, while 3Doodle and NEF are trained only from a single reference view. Naturally, multi-view supervision leads to better 3D consistency. I think a better comparison would be another baseline like this: first train a 3D representation such as NeRF using the text2image or image2image SDS loss, then use the multi-view renderings of this representation to provide multi-view supervision for training other 3D sketch reconstruction methods, followed by a comparison in terms of quality and efficiency to truly demonstrate the benefits of directly optimizing 3D curves with SDS loss.

Given above points, I believe the paper's contributions may not be substantial enough to justify its acceptance. However, I am open to raise my score if the authors can provide a convincing explaination to the above concerns.

**Questions:**

1. In some examples, such as the phoenix, the 3D curves appear insufficient to fully capture the object's shape, and the curves still seem somewhat disorganized. Can this be solved with a higher number of 3D curves?

2. The method initializes curves within a fixed-size sphere. How heavily does this initialization impact the final optimization?

3. Why not consider optimizing additional parameters, such as the width, color, or transparency of the 3D curves? Could optimizing these properties lead to significantly better results?

---

> ### Author Response · Authors · 2024-11-25
> **Response (1)**
>
> Thanks for your valuable review and suggestions. Below we provide a point-to-point response to all comments. If our response has addressed the concerns and brought new insights, we will highly appreciate it if the score can be raised.
>
> **Q1. One of the claimed novelty is the use of perspective projection with depth-aware rasterization. A similar perspective projection has already been discussed in 3Doodle (specifically in Appendix A.2), which seems to have no clear difference with the procedure proposed in this paper.**
>
> When transforming the 3D Bézier curve into 2D curves with perspective projection, the 2D projected curve is identical to a 2D rational Bézier curve defined by projected control points.
> To directly use DiffVG, which **only supports ordinary Bézier curves**, 3Doodle made a compromise by approximating the perspective projection as an orthographic projection.
> Specially, 3Doodle first projects control points to image plane with perspective projection, and then it draws **2D Bézier curve** in image plane with the projected points, which approximates the **rational Bézier curve**.
>
> To accurately formulate the differentiable rasterization of the projected 2D rational Bézier curves, we propose a novel rasterizer that supports **2D linear, quadratic, and cubic rational Bézier curves**, which cannot be achieved by 3Doodle's ordinary (not rational) Bézier curve representation.
> We have comprehensive discussed the design details in Section 4.3 and provided theoretical derivations in Appendix C.
> Figure 2 has shown the relevant results, demonstrating further control over the shape of the curves by adjusting the weights of different control points.
>
> **Q2. While fixing occlusions in 3D curves is a positive addition to the rasterizer, there is no conclusive experimental evidence showing how this improvement impacts the quality of generation. For instance, it remains unclear whether depth-aware rasterization significantly enhances the final results beyond the occlusion analysis in Figure 11.**
>
> The proposed depth-aware rasterization can mitigate the noticeable depth or color conflicts which will disrupt the visual coherence of objects and lead to spatial and semantic ambiguity.
> We have conducted additional experiments to further evaluate its effectiveness, particularly on colored 3D sketch generation.
> Please refer to the **Q2 of Common Response**.
>
> **Q3. Applying SDS loss to rasterized images is relatively straightforward and does not present a significant technical difficulty, given its widespread use in NeRF-related tasks. This loss could be easily incorporated into prior approaches without altering their core components.**
>
> Although the SDS loss has been widely used in NeRF-related tasks, it has not been investigated in the 3D sketch generation task.
> Since the rendered sketch image is highly different from the conventional image and the sketch image is generated by the differentiable curve rendering process, the optimization is also challenging and it needs careful adjustments of the SDS components to guide the sketch generation.
>
> We extensively discuss how the SDS components affect the 3D sketch results, such as the CFG weights and a biased sampling strategy, providing deep insights for future potential SDS-based sketch generation work, e.g., 3D sketch video generation and 3D sketch scene generation.
> We believe these technical designs and discussions will benefit the development of the community.

---

> ### Author Response · Authors · 2024-11-25
> **Response (2)**
>
> **Q4. Moreover, the experiments do not convincingly demonstrate the advantage of direct SDS supervision on 3D curves. I think a better comparison would be another baseline like this: first train a 3D representation such as NeRF using the text2image or image2image SDS loss, then use the multi-view renderings of this representation to provide multi-view supervision for training other 3D sketch reconstruction methods, followed by a comparison in terms of quality and efficiency to truly demonstrate the benefits of directly optimizing 3D curves with SDS loss.**
>
> As suggested, we have compared our method to the pipeline mentioned above.
> Specifically, we used MVDream and Stable-Zero123 to generate 3D objects based on the input text and a single view image, rendering 120 views (view number similar  to the NeRF Synthetic Dataset) as training data for the 3D sketch reconstruction method.
> Notably, we only report the comparison between Diff3DS and 3Doodle, as NEF struggled to converge and cannot achieve reasonable results in our experiments.
>
> For the fair comparison, 3Doodle used the same setup as ours, i.e., it only optimizes the 3D curves with randomly initialized positions of the curve control points and does not utilize the prior from the point clouds obtained from the multi-view images.
> Also, the superquadric used by 3Doodle is not considered in this experiment.
> The results have provided in **Appendix D.6 of the revised PDF**.
>
> We analyze the results from the perspectives of both quality and efficiency below.
>
> **Quality Comparison**:
>
> Overall, the quality of the sketches generated by the proposed pipeline and ours is comparable, indicating the SDS-based sketch generation can achieve similar qualitative performance comparing to the sketch reconstruction based on hundreds of multi-view images.
> Meanwhile, it should be noted due to time limit, the current comparison is not conducted on the colored sketch generation, which may reveal more differences for the two approaches, as shown in **Figure 11** and **Appendix D.7 of the revised PDF**.
>
>
> **Efficiency Comparison**:
> 1) For the Text-to-3D task, 3Doodle requires 3 hours (1 hour for 3D object generation and 2 hours for 3D sketch reconstruction), while our method takes 1 hour.
> 2) For the Image-to-3D task, 3Doodle requires 2.5 hours (0.5 hours for 3D object generation and 2 hours for 3D sketch reconstruction), while our method takes 2 hours.
>
>
> **Q5. In some examples, such as the phoenix, the 3D curves appear insufficient to fully capture the object's shape, and the curves still seem somewhat disorganized. Can this be solved with a higher number of 3D curves?**
>
> The primary reason for this issue is that the original SDS does not provide a strong structural similarity prior, in contrast to 3Doodle used losses like CLIP Loss and LPIPS Loss.
> However, for the 3D sketch generation task from text or single image input, since there is no multi-view ground truth supervision which can be used for the CLIP loss and LPIPS loss, we adopt the SDS loss for the curve optimization.
> The ablation study illustrated in Figure 9 (c) indicates that increasing the number of curves from 56 to 112 does not result in a more organized output.
> One promising solution could be to provide additional point cloud priors during initialization, similar to what 3Doodle does, by setting them as the positions of the control points.
>
> **Q6. The method initializes curves within a fixed-size sphere. How heavily does this initialization impact the final optimization?**
>
> Based on our observations, the quality of the results is generally insensitive to minor variations in the curve initialization radius.
> However, significantly decreasing the initialization radius can still degrade the quality of the results, as demonstrated in **Appendix D.4 of the revised PDF**.
> We recommend adjusting the radius to ensure the projected curve remains fully visible within the image and occupies a reasonable area.
>
> **Q7. Why not consider optimizing additional parameters, such as the width, color, or transparency of the 3D curves? Could optimizing these properties lead to significantly better results?**
>
> Our proposed differentiable rasterizer supports optimizing the control point position, color, opacity and curve width.
> To ensure a fair comparison, in our current experiments, we adhere to the baseline setup, focusing solely on optimizing the control point position.
> To address the concern, we conduct additional ablation studies to evaluate the effects of optimizing other parameters, as shown in **Appendix D.5 of the revised PDF**.
> It can be seen our pipeline can support the optimization of additional parameters including the color, opacity and width of the 3D curves.
> Meanwhile, we observe that optimizing the curve width may result in unstable outcomes.
> We hypothesize that this instability stems from the rendered image semantics being sensitive to variations in curve width, and plan to explore this issue in our future work.

---

> ### Comment · Reviewer_TMKm · 2024-11-26
>
> I want to thank the authors for providing clarifications and more experiments about the major claims of the paper, which enhances the existing results. I am raising my score to 5.
>
> Reasons for not a higher score:
> The added results still do not convincingly demonstrate the superiority of the paper's main claimed contribution—the improved rasterizer. The results in Appendix D.6 show only similar generation quality to the new baseline, with some improvements in efficiency. The benefits of rational Bézier curves and depth-awareness are minimal, even in the case of multi-view reconstruction. Moreover, the question of why the improved rasterizer is essential to the problem of sketch synthesis remains unanswered. This leaves the paper’s claimed contributions feeling somewhat fragmented. I do not mean to suggest that the improved rasterizer lacks value—it is indeed a meaningful addition to the current 3D sketch framework. However, the paper does not sufficiently explain why this component is critical for the task of 3D sketch generation.

---

> > ### Author Response · Authors · 2024-12-03
> >
> > We sincerely appreciate that our rebuttal has been recognized and the score has been kindly raised.
> >
> > First, about the results in Appendix D.6, our proposed SDS-based sketch generation method requires only a text description or a single image to achieve quality comparable to sketch reconstruction methods that depend on hundreds of posed multi-view images.
> > Additionally, our approach offers considerable improvements in efficiency, clearly highlighting its advantages.
> > At the same time, we emphasize that our SDS-based 3D sketch generation method is designed to serve as a valuable complement to the field of 3D sketch generation, rather than as a complete replacement for reconstruction-based methods like 3Doodle.
> >
> > Second, for the benefits of rational Bézier curves and depth-awareness, our rasterizer can support more precise rendering of both rational Bézier curves and Bézier curves than approximate curve rendering used in 3Doodle.
> > Therefore, it has the potential to be applied to 3D sketch generation and reconstruction tasks that require higher accuracy in curve shape, such as reconstructing a high-quality  3D sketch from multiple high-quality 2D sketches.
> >
> > Furthermore, our depth-aware rasterizer offers more accurate reproduction of occlusion relationships between curves, enabling more complex tasks such as colored sketch generation and reconstruction.
> > To illustrate this, we have uploaded a comparison video in the **revised supplementary material**. The video includes three scenes (200 frames per scene) showcasing the results of colored sketch reconstruction.
> > In the video, around timestamps 00:09s, 00:13s, and 00:25s, the center curves are incorrectly occluded by boundary curves when using 3Doodle. In contrast, our depth-aware rasterizer significantly improves occlusion handling, particularly when the results are examined continuously. These improvements are substantial rather than minimal, demonstrating the effectiveness of our approach.
> >
> > Lastly, we revised our paper to enhance the motivation and contribution for the depth-aware curve rendering and highlighting its application to colored sketch generation.
> > Please refer to the blue text in the **Introduction**, **Section 4.3** and **Conclusion** of the **revised PDF**.

---

### Official Review · Reviewer_cfcy · 2024-11-01

**Soundness:** 3
**Presentation:** 3
**Contribution:** 3
**Rating:** 8
**Confidence:** 4

**Summary:**

Nowadays, there are a lot of works conducting optimization of 3D models from 2D diffusion priors with SDS loss. Insteading of optimizing 3D models, this work aims to output 3D sketch which is also an important 3D format in related areas. To do so, a novel differatiable rendering framework is proposed. Experimental results validate the effectiveness of the proposed methods. Compared with existing differenatiable curve rendering methods, such as 3Doodle, the proposed method can maintain depth ordering which is very helpful for the opimization.

**Strengths:**

- In general, the paper is very well-presented.
- To make 3D sketch rendering differienable is important and challenge. The paper proposed a novel one which enables the maintaining of the depth ordering. I also believe such rendering framework could benefit some other tasks.
- The results seem very good. It's far better than existing ones visually.

**Weaknesses:**

There are also some issues which can be improved.
1) Disscussions on the practical usage of 3D sketches can be improved. After the first glance of this paper, my primary curiousity is from the usage of 3D sketches. In the paper, the authors just mentioned "widely used" but did not provide examples. For me, it is really hard to imagine the usage of such 3D format. When thinking this in deep, I think 3D sketch/line can be an abstract representation of 3D structure, which can be an intermediate representation for some tasks like 3D reconstruction. In this senario, the differentiable rendering will also be very important. More discussions on this are strongly encouraged to make the application senarios more clear. More specifically, I think a subsection is needed to discuss those applications of 3D sketches.

2) The evaluation is somewhat weak. As mentioned in the title and intro, view-consistency is the target of the proposed method. Howerver, there are only one user study to measure the view-consistency which is weak. The authors is encouraged to present some numeric results of view-consistency. For example, given two adjacent views, we can do a warping based on the optical flow estimation and then calculate the alignment error. If two views are consistent, the error should be minor. This metric can be compared with some baselines such as zero-123.

3) I also have a concern about the number of curves. It seems the number of curves is a parameter needs to be set manually. This will increase the burden of the users. If it is so, please include this discussion into the limitation part.

**Questions:**

No more questions.

---

> ### Author Response · Authors · 2024-11-25
>
> Thanks for your valuable review and suggestions. Below we provide a point-to-point response to all comments. We hope our response has addressed the concerns and brought new insights.
>
> **Q1. I think 3D sketch/line can be an abstract representation of 3D structure, which can be an intermediate representation for some tasks like 3D reconstruction. In this senario, the differentiable rendering will also be very important. More discussions on this are strongly encouraged to make the application senarios more clear.**
>
> Thanks for the suggestions.
> We have included the following discussion in the **introduction section of the revised PDF (L58-L61)**:
> "Moreover, in the area of wire art generation, 3D sketch or 3D curves are also widely studied to abstract the desired visual concepts from diverse inputs, e.g., 3D surfaces (Yang et al., 2021), multi-view images (Liu et al., 2017) and text (Tojo et al., 2024; Qu et al., 2024). These representations provide essential shape and structure of the artwork as well as hold significant potential as intermediate formats for tasks such as 3D reconstruction. However, generating view-consistent 3D sketches from flexible inputs, such as text or a single image, remains largely unexplored."
>
> **Q2. The authors is encouraged to present some numeric results of view-consistency.**
>
> Thanks for the suggestion. Currently, the view-consistency has been evaluated by the user study which asked the user to rate the four provided views based on the criteria including the 3D consistency. It can also be examined from the videos provided in the supplementary material.
> To further evaluate the view-consistency in a quantitative manner such as utilizing the optical flow to generate the warped images is very interesting and we will explore this direction in our future work.
>
> **Q3. It seems the number of curves is a parameter needs to be set manually. If it is so, please include this discussion into the limitation part.**
>
>  Yes, the number of initialized curves needs to be specified by the user.
>  We have provided the results of different initial curve numbers in Figure 9(c).
>  Based on the experiments, we set the curve number to be 56 to achieve a balance between the approximation accuracy and complexity of the results.
>  We have included the discussion on this limitation in the **conclusion section of the revised PDF**:
>  "Also, the initial curve number is set manually to achieve a balance between the approximation accuracy and complexity of the results."

---

### Official Review · Reviewer_Jp4d · 2024-11-03

**Soundness:** 3
**Presentation:** 3
**Contribution:** 2
**Rating:** 6
**Confidence:** 3

**Summary:**

The paper proposes a framework for generating 3D sketches from either a text prompt or a single image. It is based on differentiable rendering of 3D Bézier curves and applies supervision from pretrained image diffusion models from random viewpoints.

**Strengths:**

* Covers multiple aspects of differentiable rendering.
* Extends DiffVG, a differentiable rasterizer for 2D vector graphics, to 3D Bézier curves.
* Presents a relatively simple and elegant pipeline, clearly summarized in Figure 3.
* Includes small but useful techniques that are well explained and analyzed:
    * Time annealing for the Score Distillation Sampling (SDS).
    * Curve filtering to reduce noise.

**Weaknesses:**

* The z-ordering of Bézier curves in Section 4.3 is the most technically advanced contribution. But the motivation is unclear: all sketches consist of black strokes, and intersections are rare at the image scale. Why care about correctly ordering the curves? Figure 11’s ablation shows only very minor visual differences. The effect of this ordering on 3D sketch generation is also left unclear.
* In the related works section, the motivations against existing 3Doodle approaches seem weak:
    * Issues with perspective projections.
    * Use of z-ordering for handling colored intersections.
* The usefulness and relevance of 3D sketches as shape representation is not fully clarified.

**Questions:**

* Why care about z-ordering of the curves? In which situation is it relevant?
* Figure 4: The meaning is unclear. Is this supposed to represent a pseudo-ground truth image? How does it relate to gradient supervision?
* In which situation are 3D sketches useful?

---

> ### Author Response · Authors · 2024-11-25
>
> Thanks for your valuable review and suggestions. Below we provide a point-to-point response to all comments. If our response has addressed the concerns and brought new insights, we will highly appreciate it if the score can be raised.
>
> **Q1. The z-ordering of Bézier curves in Section 4.3 is the most technically advanced contribution. But the motivation is unclear: all sketches consist of black strokes, and intersections are rare at the image scale. Why care about correctly ordering the curves? Figure 11’s ablation shows only very minor visual differences. The effect of this ordering on 3D sketch generation is also left unclear. Why care about z-ordering of the curves? In which situation is it relevant?**
>
> First, we would like to clarify our motivation on introducing the depth-aware rasterization for rational Bézier curve rendering. Research on reconstructing or generating 3D sketches using multimodal pre-trained models is an emerging but underexplored area.
> The state-of-the-art method, 3Doodle, integrates DiffVG, a differentiable rasterizer originally designed for 2D vector graphics, as a component for rendering 3D curves.
> However, its performance is constrained by the inherent limitations of DiffVG's original design.
> To overcome this issue, we introduce a depth-aware rasterizer that enable precise and differentiable rendering of both black and colored 3D curves.
> We aim to advance research and foster further development in the direction of both 3D sketch reconstruction and generation.
>
> In the current experimental setup, all curve colors are fixed to black to ensure fair comparison with baseline methods.
> Under this condition, depth-based rasterization indeed has minimal impact on the final results.
> On the other hand, z-ordering is crucial for more complex tasks like colored sketch generation.
> Accurate z-ordering ensures that occlusion relationships between curves are faithfully reproduced, enabling precise spatial and semantic representation.
> We conduct additional experiments to evaluate its effectiveness.
> Please refer to the **Q2 of Common Response**.
>
> **Q2. In the related works section, the motivations against existing 3Doodle approaches seem weak: Issues with perspective projections. Use of z-ordering for handling colored intersections.**
>
> Thanks for pointing out this issue. We will enhance the motivation against 3Doodle in the related work section. Specifically, in addition to the technical aspects like the perspective projection and z-ordering, we focus on using the simple text or single image input to achieve 3D sketch generation, while 3Doodle requires hundreds of rendered images for 3D sketch reconstruction.
> Although combining 3Doodle with some SOTA text-to-3D methods would achieve similar sketch generation results, our method is more efficient (as demonstrated in **Appendix D.6 of the revised PDF**), and it leverages end-to-end SDS-based differentiable curve optimization without relying on additional text-to-3D generation models.
>
>
> **Q3. The usefulness and relevance of 3D sketches as shape representation is not fully clarified. In which situation are 3D sketches useful?**
>
> Sketching is a powerful tool for visualizing concepts and ideas.
> 3D sketch which abstracts the 3D shape into 3D curves is a novel 3D representation that can provide more view-variant information than traditional 2D sketches.
> One key application of 3D sketches is the generation of multi-view consistent 2D sketches from different perspectives, facilitating a more comprehensive understanding of visual concepts.
> Additionally, this representation has great potential to serve as an intermediate representation for other tasks such as 3D reconstruction.
>
> **Q4. Figure 4: The meaning is unclear. Is this supposed to represent a pseudo-ground truth image? How does it relate to gradient supervision?**
>
> According to Equations 11 to 14, the SDS loss, which is conventionally formulated as the gradient form, can be interpreted as a scaled L2 loss that compares the rendered view with the pseudo-ground truth image predicted by the pre-trained text-to-image model.
> The differences are then used to update the 3D curve parameters through back-propagated gradients.
>
> Figure 4 illustrates the pseudo-ground truth images under different levels of CFG weight, which is a key hyper-parameter in the SDS-based image or 3D generation.
> A larger weight provides a stronger shape supervision prior, while a smaller weight leads to shape degradation.
> The relevant ablation results are presented in Figure 9(a).

---

### Official Review · Reviewer_w44R · 2024-11-04

**Soundness:** 3
**Presentation:** 4
**Contribution:** 3
**Rating:** 6
**Confidence:** 4

**Summary:**

The authors present Diff3DS, a framework for optimizing 3D strokes using text or image guidance. This work introduces a novel extension of DiffVG that allows users to differentiably rasterize 2D projections of 3D parametric curves with more accurate depths / occlusions based on the 3D location of the curves. Using this new rasterization technique, Diff3DS applies SDS supervision to generate view-consistent 3D sketches using text or image guidance. This work validates its approach through ablations of key components and comparisons to other similar works. Diff3DS shows improved performance over other methods in both automated metrics and a perceptual user study. The paper is well written and easy to follow. The method is explained in sufficient detail and design decisions are properly motivated. Informative figures clearly showcase the contributions of this work (i.e. Figure 2 nicely demonstrates this ability to have non-uniform ordering between curves). The provided experiments validate claims made in the paper.

**Strengths:**

Strengths:
- Through the use of score distillation sampling for supervision, this work is the first to enable text/image driven 3D stroke generation from scratch (as opposed to existing works which simplify / abstract / sketchify an existing 3D representation defined by multi-veiw images).
- The novel differentiable rasterization technique introduced by this work allows for more accurate depths when rasterizing 3D curves.
- The paper is well written and organized.

**Weaknesses:**

Weaknesses:
- The main contribution of this work appears to be the improved differentiable rasterization approach that enables more accurate depths during rasterization. However, it is not clear that this component is responsible for improved performance. Figure 11 ablates the impact of this rasterization technique on 3Doodle. However, the differences are quite minor and at this level of abstraction, the depths of these curves seem less important to the visual coherence of the object.
- The comparisons currently shown do not use strong enough / similar enough baselines. Some of the comparisons shown are to standard (non-sketch) generative approaches. Showing these is fine, especially given the lack of existing work on text driven 3D consistent sketch generation. However, it feels that using score distillation guidance + DiffVG should still produce strong results for text/image guided curve generation. This is an important baseline and should be compared to in the paper.

**Questions:**

It appears from Figure 6 that, while your approach produces more 3D consistent results than 3Doodle, it less closely captures the structure of the guiding image. Do you have a sense of why that might be? Is it due to the fact that Diff3DS is more 3D consistent and thus is less able to overfit to exact structural details? Can this be improved?

---

> ### Author Response · Authors · 2024-11-25
>
> Thanks for your valuable review and suggestions. Below we provide a point-to-point response to all comments. If our response has addressed the concerns and brought new insights, we will highly appreciate it if the score can be raised.
>
> **Q1. The main contribution of this work appears to be the improved differentiable rasterization approach that enables more accurate depths during rasterization. However, it is not clear that this component is responsible for improved performance. Figure 11 ablates the impact of this rasterization technique on 3Doodle. However, the differences are quite minor and at this level of abstraction, the depths of these curves seem less important to the visual coherence of the object.**
>
> First, please refer to **Q1 of Common Response** for our main contributions.
> For the rasterizer part, our contributions include:
>   1) Extending DiffVG to support the differentiable rasterization of rational Bézier curves, enabling precise 3D curve perspective projection consistent with theoretical derivations.
>   2) Introducing depth-based rasterization, which facilitates accurate color computation and faithfully reproduces occlusion relationships between 3D curves in space.
>
> In the current experimental setup, to ensure a fair comparison with baseline methods, all curve colors were fixed to black.
> Under this condition, depth-based rasterization had a minimal impact on the final results.
>
> Nonetheless, our rasterizer shows considerable potential for tasks involving the rendering or generation of colored sketches.
> Notably, noticeable depth or color conflicts would disrupt the visual coherence of objects and lead to spatial and semantic ambiguity.
> We conduct additional experiments to evaluate its effectiveness.
> Please refer to the **Q2 of Common Response**.
>
> **Q2. The comparisons currently shown do not use strong enough / similar enough baselines. It feels that using score distillation guidance + DiffVG should still produce strong results for text/image guided curve generation. This is an important baseline and should be compared to in the paper.**
>
> As mentioned above, DiffVG, which is designed for differentiable rendering of 2D curves, is not inherently suitable for the theoretical framework of 3D curve rendering.
> A significant contribution of our work is the adaptation and extension of DiffVG to support tasks that require accurate perspective projection and occlusion handling for **3D curves**.
>
> To address the concern about the lack of sufficiently similar baselines, we add a new baseline method: first, we train a 3D representation use the SDS algorithm guided by text or image, and then use the multi-view renderings for training 3D sketch reconstruction methods.
>
> We compare with the new baseline on both text-guided and image-guided 3D sketch generation, with results detailed in **Appendix D.6 of the revised PDF**.
> Across all tasks, we achieve performance comparable to the baseline method while considerably reducing the training time.
>
> **Q3. It appears from Figure 6 that, while your approach produces more 3D consistent results than 3Doodle, it less closely captures the structure of the guiding image. Do you have a sense of why that might be? Is it due to the fact that Diff3DS is more 3D consistent and thus is less able to overfit to exact structural details? Can this be improved?**
>
> The primary reason for this issue lies in the absence of a strong structural similarity prior in the original SDS method.
> In contrast, 3Doodle leverages loss functions such as CLIP Loss and LPIPS Loss, which effectively promote structural similarity.
> However, for the 3D sketch generation task from text or single image input, since there is no multi-view ground truth supervision which can be used for the CLIP loss and LPIPS loss, we adopt the SDS loss for the curve optimization.
> A potential solution could involve introducing additional point cloud priors during curve initialization, similar to 3Doodle’s approach, by setting them as control point positions.
> Moreover, as mentioned in the limitation part, distinguishing between view-independent curves (e.g., feature lines) and view-dependent curves (e.g., contours of smooth surface boundaries) could further enhance the expressive capacity of a 3D object’s overall shape.

---

> > ### Comment · Reviewer_w44R · 2024-11-26
> >
> > I thank the authors for their thorough response. My biggest concern with the original submission was that the technical contribution of this depth-aware 3D curve rendering does not clearly improve the overall result. Figure 11 and Supplementary section D.7 in the revised submission do a better job at demonstrating this than in the original submission, specifically for the application of using colored curves for object/scene abstraction. The example with the hotdog being occluded by the plate makes this point especially clear. However, this colorized abstraction task is a different application than general sketch abstractions which are typically uncolored. In this more general, uncolored case, it is still not clear that the proposed technical contribution improves the quality since occlusions are not that noticeable. In light of the new figure 11 and section D.7, I am raising my score to a 6, but still think this submission would benefit from re-framing its main task to colored sketch abstraction where the technical contributions have more effect on the result.

---

> > > ### Author Response · Authors · 2024-12-03
> > >
> > > We sincerely appreciate that our rebuttal has been recognized and the score has been kindly raised.
> > > First, we agree that the proposed depth-aware curve rendering method is more effective for generating colored sketches, a task that cannot be adequately addressed by previous rasterizers like DiffVG-based 3Doodle.
> > > Re-framing the main task as colored sketch generation is a very good suggestion.
> > > However, due to time constraints, we have focused our revisions on enhancing the motivation behind depth-aware curve rendering and highlighting its application to colored sketch generation.
> > > Please refer to the blue text in the **Introduction**, **Section 4.3** and **Conclusion** of the **revised PDF**.
> > > We believe it will be a promising direction for future work to fully apply and comprehensively evaluate our method for colored sketch generation.

---

### Author Response · Authors · 2024-11-25
**Common Response**

We sincerely appreciate the constructive comments from all reviewers to further improve our paper.
We find there are common concerns to our paper, and we would like to clarify them here.

**Q1. Novelty and contribution**

First, we would like to re-emphasize the novelty and contribution of our work.
In addition to the improved differentiable rasterization mentioned by reviewers, we introduce the first effective and efficient end-to-end 3D sketch generation pipeline, enabling broader applications beyond 3D sketch reconstruction, such as those supported by 3Doodle.
Specifically, our key contributions are as follows:

1) We present the first end-to-end framework for 3D sketch generation, supporting flexible text to 3D sketch and single image to 3D sketch generation.

2) We extend DiffVG to support the differentiable rasterization of rational Bézier curves, enabling accurate 3D curve perspective projections consistent with theoretical formulations.

3) We propose depth-based rasterization, enabling precise color computation and faithfully capturing occlusion relationships between 3D curves, thereby enhancing both visual coherence and semantic accuracy.

**Q2. The effect of our proposed depth-aware rasterizer**

Our rasterizer accurately renders the blended colors in overlapping regions and faithfully reproduces occlusion relationships between curves, showcasing its strong potential for generating colored 3D sketches.
To assess its effectiveness, we conducted an experiment to generate colored 3D sketches from multi-view images using both the original 3Doodle and a variant integrated with our rasterizer.
We utilized the object dataset provided by 3Doodle, as well as the NeRF Synthetic dataset for the evaluation.

As illustrated in updated **Fig. 11 of the revised PDF**, the original 3Doodle suffers from noticeable color conflict errors across multiple viewpoints (e.g., the support base, which is farther from the camera, is incorrectly rendered in the foreground, occluding the LEGO excavator).
These noticeable color conflicts disrupt the object’s visual coherence and lead to semantic ambiguity.
In contrast, our rasterizer consistently reproduces accurate occlusion relationships between objects, providing clearer visual and spatial semantics.

Additional results are provided in **Appendix D.7 of the revised PDF**.

---

### Meta-Review · Area_Chair_vWdA · 2024-12-22

**Metareview:**

This paper introduces an approach for 3D sketch generation. The core idea is to propose a new differentiable rendering framework for generating view-consistent 3D sketches, which involves perspective projections to render 3D Bezier curves into the 2D image domain. The paper utilized score distillation sampling (SDS) to generate 3D sketches from texts and images. AC confirms that this paper introduces interesting contributions regarding the Bezier curves perspective rasterizer to improve 3D sketch rendering quality, and the successful adaptation to the 3D sketch generation task can shed light on the field.

**Additional Comments On Reviewer Discussion:**

All reviewers recommend paper acceptance after the rebuttal phase. AC confirms that the authors provided solid feedback on the reviewers' comments. In particular, the reviewer TMKm raised the core after the discussion phase. The common concerns raised by reviewers were mainly about technical novelty and the effect of the proposed depth-aware rasterizer. The authors provided extra results in the revised PDF file.

Specifically, reviewer w44R asks about the technical contribution and weak comparison results with other approaches, such as 3Doodle. Given the detailed feedback, the reviewer raised the score to 6 but suggested reframing the paper as the colored sketch abstraction. The reviewer Jp4d questioned the motivation of the depth-aware rasterizer, the less impressive comparison with the 3Doodle, and the usefulness of the 3D sketches. The authors provided feedback on this, and the reviewer stated that the paper offered valuable insights during the AC-reviewer discussion phase. The reviewer cfcy scored strongly and requested clarification of the application scenarios, quantitative view consistency, and manual parameters. After reviewing the authors' feedback, the reviewer stated that the original score should be kept in the AC-reviewer discussion phase. The reviewer TMKm, who provided the most detailed comment, was skeptical about technical novelty and unclear effects from the aware rasterizer, and the reviewer thinks that applying SDS loss is a straightforward extension. The other questions regarding uncompelling results have still not been resolved after the author's rebuttal. During the reviewer-AC discussion phase, the reviewer TMKm states that the reviewer agrees with the comment by Jp4d regarding the comment that this paper is the first to formulate the task of the 3D generation, and the reviewer TMKM raised the score to 6.

AC confirms that the discussion or suggestion was constructive, and the revised version is much better after the rebuttal phase.

---

### Decision · Program_Chairs · 2025-01-22

Accept (Poster)